# TEST ERROR GUARANTEES FOR BATCH-NORMALIZED SHALLOW RELU NETWORKS TRAINED WITH GRADIENT DESCENT

## ABSTRACT

This work establishes low training and test error guarantees of gradient descent (GD) and stochastic gradient descent (SGD) on two-layer ReLU networks with Batch Norm. Prior work provided convergence analyses for low training error or stationary points while critically relying on modifications to the setting such as modifying Batch Norm and assuming the objective is smooth. Although smoothness based analyses can handle deeper networks, the smoothness constants are highly non-negligible. We take an alternative approach using a margin $\gamma$ tailored to normalized networks. In particular, for a network of width $m$, the test errors for GD and SGD decrease at a rate of $\mathcal{O}(\frac{m^{1/3}}{\gamma^{1/3}t})$ and $\mathcal{O}(\frac{1}{\gamma^2 t})$ up until $t \approx O(\frac{\exp(\gamma^2 m)}{n})$. Along the way, we show that $\gamma$ can be $O(\sqrt{d})$ times larger than the margin of the max margin linear predictor which can potentially explain the training and test error speed up for normalized networks.

## 1 INTRODUCTION

While deep network architectures undergo constant evolution, few of these constant changes are so beneficial that they persist in *all* future architectures. One such design decision is the pervasive use of *normalization layers*, foremost amongst them *batch normalization (BN)* (Ioffe & Szegedy, 2015). As mentioned, all modern architectures feature some sort of normalization layer, such as LLaMa (Touvron et al., 2023), which uses RMSnorm (Zhang & Sennrich, 2019), and a simple normalization choice even appears in classical architectures (Fukushima & Miyake, 1982).

The immediate widespread adoption of BN led to an intense scrutiny of its benefits; following initial questions about its motivation for *reducing covariate shift* (Rahimi & Recht, 2017), a variety of authors studied various aspects empirically, for instance negating the covariate shift hypothesis but positing other benefits such as making the loss surface smoother (Santurkar et al., 2018), and empirical support of its optimization and generalization benefits (Bjorck et al., 2018). Despite all the efforts, there lacks analysis for loss convergence and test error for nonlinear networks with normalization layers trained with standard descent methods.

In the classification setting, Kohler et al. (2019) showed convergence to stationary points for two layer networks with tanh activation. Arora et al. (2018) also show convergence to stationary points for multi-layer networks trained using GD but uses a modified BN layer which avoids a technical hurdle that we deal with in this paper. In particular, they scale the nonzero constant in batch normalization (typically used in practice to avoid dividing by zero errors) by the norm of the incoming node. While seemingly innocous, this modification ensures that the norms of the layers being normalized grow monotonically which in turn ensures that the reciprocal of the normalization factors in batch norm are well-behaved. Furthermore, in prior work, gradients are assumed to be lipschitz which is problematic as the lipschitz constants can be as large as $O(M^L)$ where $M$ is the max width of the network and $L$ is the total number of layers and thus cannot be ignored. In this paper, we instead employ margin based techniques utilized in Telgarsky (2022) that also establish low training and test error but for two-layer networks *without* normalization.

## 1.1 CONTRIBUTIONS

This work establishes convergence rates for loss and test error bounds that scale inversely with a margin like quantity. Throughout the rest of the paper, let $\gamma$ be the margin for networks with batch norm, fully defined in section 1.4.

1. **Section 1.4: Margin for normalized networks.** We derive a notion of margin tailored to networks with normalization. Broadly speaking, we use nonlinear data separators and margins to construct a useful reference parameter. Finding a good reference parameter is key to making analysis of the standard Euclidean potential and variants of thereof tractable which in turn grants training and test error bounds. While our notion of margin is unique, the margin based techniques used to prove theorem 2.1 and theorem 2.2 are derived from Telgarsky (2022); Ji & Telgarsky (2020). Furthermore, we utilize a rescaled Euclidean potential to prove theorem 2.1 and theorem 2.2 in order to handle layers of different degrees of homogeneity which may be of independent interest.

2. **Section 2: Gradient Descent, Stochastic Gradient Descent, and potential functions.**

   (a) **Training and test error for GD on batch-normalized 2-layer networks.** Theorem 2.2 establishes convergence rates for loss and test error bounds for GD. Specifically, GD obtains $\epsilon > 0$ test error after $\mathcal{O}(\frac{\gamma^{1/3}m^{2/3}}{\epsilon\sqrt{n}})$ iterations while only needing the width of the network to be $\tilde{\mathcal{O}}(\frac{1}{\gamma^2})$. In contrast, prior results fall under the following categories: convergence rates to global minimizer of loss or to stationary points. Results of the former only applies to *linear networks* with batch normalization (Kohler et al., 2019; Cai et al., 2019). For the latter category, Kohler et al. (2019) also proves convergence rates to critical points for shallow networks but they freeze the outer layer, train Batch Norm parameters using a bisection method, and apply gradient descent to *unnormalized network* and normalize the parameters after training is complete. Similarly, Arora et al. (2018) also obtains convergence rates to critical points for deep networks but they assume bounds on smoothness which can be as large as $\tilde{\mathcal{O}}(M^L)$ where $M$ is the max width of the network and $L$ is the depth of the network. Furthermore, they modify Batch Norm to ensure that the network is smooth and scalar invariant with respect to the normalized layer. Note, in practice this scalar invariance fails to hold as a nonzero $\epsilon$ is added to the variance term in Batch Norm. While seemingly negligible, lack of scalar invariance allows the normalization factor in Batch Norm to shrink which in turn allows the norm of the gradients to explode. In particular, this breaks the analysis used in Arora et al. (2018).

   (b) **Train and test error for SGD on batch-normalized 2-layer networks.** Theorem 2.1 establishes convergence rates for loss and test error bounds for SGD. In the case of SGD, we require $\tilde{\mathcal{O}}(\frac{1}{\epsilon\gamma^2})$ iterations to get $\epsilon > 0$ test error while only needing the width of the network to be $\tilde{\mathcal{O}}(\frac{1}{\gamma^2})$. One caveat is the usage of the mean and variance statistics of the whole dataset, rather than using minibatch statistics as in practice, which we leave to future work. As with GD, Arora et al. (2018) also obtains convergence rates to critical points for deep networks though assuming bounds on smoothness and modifying Batch Norm as mentioned above.

   (c) **Potential functions.** We utilize a rescaled Euclidean potential to prove theorem 2.1 and theorem 2.2 in order to handle layers of different degrees of homogeneity which may be of independent interest.

## 1.2 RELATED WORK

**Sources of BN optimization advantage** Many explanations have been proposed for why batch normalization improves optimization speed, with the original proposal being its mitigation of "internal covariate shift" (Ioffe & Szegedy, 2015). Later work attempts to analyze BN empirically and theoretically, with many of the explanations roughly divided into the following two categories.

**Gradient and activation norms.** Early empirical analysis of BN (Bjorck et al., 2018) considers how BN addresses the well-established problem of exploding gradients (Hochreiter & Schmidhuber, 1997; He et al., 2016; Philipp et al., 2017). They demonstrate that networks with batch normalization

are more stable since the activation and gradient distributions of BN networks are significantly less heavy-tailed than networks without BN; this stability allows larger learning rates and resulting in faster training as demonstrated in other work (Hanin & Rolnick, 2018). Further work (Lubana et al., 2021) generalizes the link between the size of gradient norm and training stability, demonstrating that normalization layers that can encourage large gradient norms, such as Instance Norm (Ulyanov et al., 2016), tend to lose their stability benefits.

**Optimization landscape smoothness.** As an alternative to the internal covariate shift hypothesis, works have suggested that BN improves the smoothness of the optimization landscape (Santurkar et al., 2018; Karakida et al., 2019). Karakida et al. (2019) characterizes a notion of landscape smoothness and proves that applying BN on the last layer can significantly reduce the sharpness of the loss surface (Karakida et al., 2019). Studying the application of BN only on the last layer can be justified by Bjorck et al. (2018) which empirically shows batch normalization on the final layer provides the bulk of the benefits that BN is commonly known to provide (Bjorck et al., 2018). Further theoretical analysis indicates that other forms of normalization, such as weight normalization (Salimans & Kingma, 2016), also yield favorable effects on the optimization landscape (Dukler et al., 2020). In addition, Lyu et al. (2022) provably show that for scale-invariant losses, gradient descent coupled with weight decay implicitly obtain solutions that have small spherical sharpness, a quantity defined as the maximal eigenvalue of the loss Hessian over the sphere.

**Convergence analyses.** Wu et al. (2018) analyzes convergence to a critical point for smooth functions using an adaptive method inspired by weight normalization. Arora et al. (2018) builds upon techniques used in Wu et al. (2018) to establish convergence rates to critical points for smooth networks that use a modification of batch normalization. In the realm of regression, (Cai et al., 2019; Kohler et al., 2019) also analyze convergence to critical points for batch-normalized linear models where the linear layer is trained using GD and the affine parameter is trained via a bisection method. For weight normalization, Dukler et al. (2020) prove a training error guarantee for weight-normalized two layer networks using square loss where the outermost layer is fixed and only the affine and inner layer are trained.

**Replicating benefits induced by BN without using BN.** A line of work aims to recreate the benefits of BN without actually using normalization (Zhang et al., 2019; De & Smith, 2020; Shao et al., 2020; Brock et al., 2021a) such as Fixup Initialization (Zhang et al., 2019), which recreates the optimization benefits of BN even on deep networks like ResNet, and yields near-SOTA results on ImageNet without normalization. Interestingly, they report that Fixup Initialization overfits more strongly than BN providing more support for BN regularization effects. More recent work synthesizes a fairly broad number of proposed intermediate factors contributing to the advantages provided by BN, yielding more effective methods which replicate and improve upon BN advantages without using normalization (Brock et al., 2021b). In particular, the authors propose a normalization-free variant of ResNets called NFNets, which attain higher test accuracy and are more computationally efficient than traditional BN ResNets.

## 1.3 NOTATION

Before stating our main results, we first develop some notation.

**Problem setting and norms.** We will work in the binary classification setting where we are given a dataset $\{(x_i, y_i)\}_{i \in [n]}$ and $y_i \in \{-1, +1\}$. Let $\mu := \frac{1}{n} \sum_{i=1}^{n} x_i$ and $\Sigma := \frac{1}{n} \sum_{i=1}^{n} (x_i - \mu)(x_i - \mu)^\top$. We assume that the mean centered data points $\overline{x_i} = x_i - \mu$ satisfy $\|\overline{x_i}\| \leq 1$. Unmarked norms $\|\cdot\|$ are $\ell_2$ norms. Let $\|\cdot\|_\Sigma$ denote the seminorm such that $\|v\|_\Sigma = \sqrt{v^\top \Sigma v}$. Fix $\epsilon \geq 0$ and define the batch normalization factor as $N(v) = \sqrt{\|v\|_\Sigma^2 + \epsilon}$.

**Initialization and architecture.** We shall use roughly the pytorch default initialization schemes, meaning $a \sim \mathcal{N}_m / \sqrt{m}$ (vector whose entries are i.i.d gaussian with variance $\frac{1}{m}$) $V \sim \frac{\mathcal{N}_{m \times d}}{\sqrt{d}}$ ($m \times d$ matrix whose entries are i.i.d gaussian with variance $\frac{1}{d}$) and $c = \vec{\mathbf{1}} \in \mathbb{R}^m$. We will work with shallow networks with batch norm $f(x; W = (a, c, V)) = \sum_{j \in [m]} a_j \sigma \left( c_j \left\langle \frac{v_j}{N(v_j)}, x - \mu \right\rangle \right)$.

**Derivatives and suppressed notation.** To reduce notational clutter, we often suppress the contents of $\sigma'$ as follows $o_j|_{\overline{x_i}} := \sigma'\left(c_j\left\langle\frac{v_j}{N(v_j)}, x-\mu\right\rangle\right)$. We will often group the outer and affine parameters together as $U = (a, c)$. For the reader's convienience, we provide the partial derivatives:

$$\partial_a f(W; x) = \sum_{j\in[m]} \sigma\left(c_j\left\langle\frac{v_j}{N(v_j)}, x-\mu\right\rangle\right) e_j,$$

$$\partial_c f(W; x) = \sum_{j\in[m]} a_j o_j|_{\overline{x_i}}\left\langle\frac{v_j}{N(v_j)}, x-\mu\right\rangle e_j,$$

$$\partial_{v_j} f(W; x) = \frac{a_j c_j o_j|_{\overline{x_i}}}{N(v_j)}\left(I - \frac{\Sigma v_j v_j^T}{N(v_j)^2}\right)\overline{x_i}.$$

We also make the following data assumption.

**Assumption 1.1.** *For $j \in [m]$, let $v_j \sim \frac{\mathcal{N}_d}{\sqrt{d}}$. With probability at least $1 - \delta$, we have*

$$\max_j \frac{1}{N(v_j)} \leq \epsilon_N. \tag{1}$$

When $\epsilon > 0$, we trivially have that with probability 1, assumption 1.1 is satisfied for $\epsilon_N = \frac{1}{\sqrt{\epsilon}}$

**Stochastic gradient descent (SGD) and gradient descent (GD):** We will use the logistic loss $\ell(z) = \ln(1 + \exp(-z))$. We also write $\ell_i(W) = \ell(p_i(W)) = \ell(y_i f(x_i; W))$ and the emprical risk as $\widehat{\mathcal{R}}(W) := \frac{1}{n}\sum_{i\in[n]}\ell_i(W)$. The update for SGD is

$$W_{i+1} := W_i - \eta\partial_W\ell_i(W) \qquad\qquad \text{SGD},$$

whereas GD is given as the solution to a differential *inclusion*, specifically

$$W_t := \dot{W}_t = -\partial_W\widehat{\mathcal{R}}(W_t) \qquad\qquad \text{GD}.$$

For both SGD and GD, we let $\partial$ denote an element of the Clarke differential. For more details on these nonsmooth derivative formalism once again the reader is directed to the excellent work of Clarke (1975), whose differential notion is now standard in the analysis of ReLU networks.

We open the section with the derivation of the margin for batch normalized networks and a computation of the margin on a specific dataset where the Batch Norm margin is $\mathcal{O}(\sqrt{d})$ larger than the margin of the max margin linear predictor (section 1.4). In section 2, we state formal theorems about test error guarantees for normalized networks trained with SGD and GD.

### 1.4 MARGIN FOR BATCH-NORMALIZED NETWORKS

This section is devoted to motivating the usage of the margin $\gamma$ that is tailored to Batch Norm. In optimization, a common potential is the standard Euclidean potential. Concretely, if we have some iterative algorithm that generates weights $\{W_s\}_{s\in[t]}$ and we are given a reference parameter $\overline{W}$, the squared Euclidean potential is

$$\left\|W_0 - \overline{W}\right\|^2 - \left\|W_t - \overline{W}\right\|^2.$$

To illustrate its usage, consider running GD on a convex, $\beta$-smooth objective $h : \mathbb{R}^d \to \mathbb{R}$ with step size $\frac{1}{\beta}$ for $t$ iterations, producing $t$ iterates $\{z_s\}_{s\in[t]}$. Suppose we are given a reference parameter $\overline{z}\mathbb{R}^d$. It is well known that

$$h(z_t) - h(\overline{z}) \leq \frac{\beta}{2t}\left(\|z_0 - \overline{z}\|^2 - \|z_t - \overline{z}\|^2\right).$$

Thus, it is evident that controlling the Euclidean potential controls the suboptimality gap which highlights the usefulness of the squared Euclidean potential. For technical reasons, we shall consider the *negated* rescaled squared Euclidean potential in this work:

$$\Phi(t) := \left\| U_t - \overline{U} \right\|^2 - \left\| U_0 - \overline{U} \right\|^2 + \alpha \left( \left\| V_t - \overline{V} \right\|^2 - \left\| V_0 - \overline{V} \right\|^2 \right).$$

where $\alpha > 0$ is some scaling constant. Unlike the convex optimization problem described above, the choice of $\overline{W}$ matters in our setting and is the first technical hurdle in our analysis. Since the empirical risk $\widehat{\mathcal{R}}$ is sum of exponential tailed losses, solutions are off at infinity. Therefore, a natural way of choosing reference parameter $\overline{W}$ is by considering some linear combination of our initial parameter $W_0$ and some positive margin direction $\overrightarrow{W}$. To be more explicit, let us first define the margin function

$$\gamma(W) := \min_{i \in [n]} \frac{y_i f(x_i; W)}{\|a\| \|c\|}$$

The careful reader will note that we need to also normalize by $\|V\|$ for $\gamma$ to be a true margin function if the constant $\epsilon$ in the normalization factor is nonzero. However, this detail is irrelevant as we will see soon. As mentioned before, given some scaling constant $r \geq 0$ and direction $\overrightarrow{W}$ which satisfies $\gamma(\overrightarrow{W}) > 0$, it is natural to select the reference parameter

$$\overline{W} = W_0 + r\overrightarrow{W}.$$

Unfortunately, this is rather difficult to use. Instead, we adopt the approaches in Telgarsky (2022), Nitanda & Suzuki (2019), Ji & Telgarsky (2020) and consider letting $f$ be an NTK predictor. Explicitly, we redefine the margin function such that instead of using $f$ that is a shallow network with batch normalization it is instead an NTK predictor with batch normalization: $f \in \tilde{\mathcal{F}} := \{x, W \to \langle W, \partial f(x; W_0) \rangle\}$.

Thus, given some scaling constant $r \geq 0$ and direction $\overrightarrow{W}$ which satisfies $\gamma(\overrightarrow{W}) > 0$, we again consider reference parameter

$$\overline{W} = W_0 + r\overrightarrow{W}.$$

Since each layer is initialized with different norms ($\|a\|$ concentrates around $1$, $\|c\| = \sqrt{m}$, and $\|V\|$ concentrates around $\sqrt{\frac{m}{d}}$), it is useful to split the NTK predictor into three smaller NTK predictors. In particular, we consider the decompositon

$$\langle W, \partial f(x; W_0) \rangle = \langle a, \partial_a f(x; W_0) \rangle + \langle c, \partial_c f(x; W_0) \rangle + \langle V, \partial_V f(x; W_0) \rangle$$

which leads to three natural margin functions

$$\gamma_a(a) := \min_{i \in [n]} y_i \frac{\langle a, \partial_a f(x_i; W_0) \rangle}{\|a\|},$$

$$\gamma_c(c) := \min_{i \in [n]} y_i \frac{\langle c, \partial_c f(x_i; W_0) \rangle}{\|c\|},$$

$$\gamma_V(V) := \min_{i \in [n]} y_i \langle V, \partial_V f(x_i; W_0) \rangle.$$

This decomposition is nice as we can consider different scalings for different layers when constructing the reference parameter as shown below:

$$\overline{W} := (a_0 + r_a \overrightarrow{a}, c_0 + r_c \overrightarrow{C}, V_0 + r_V \overrightarrow{V})$$

where $\overrightarrow{a_j}, \overrightarrow{c_j}, \overrightarrow{V}$ satisfy

$$\gamma := \gamma_a(\overrightarrow{a}) + \gamma_c(\overrightarrow{C}) + \gamma_V(\overrightarrow{V}) > 0.$$

We make one final modification. To account for the fact that our model is essentially 0-homogeneous with respect to $V$, we drop $\gamma_V(V)$ and instead bake in the contribution of $V$ to $\gamma$ by evaluating the inner products $\langle a, \partial_a(\cdot) \rangle$ and $\langle c, \partial_c(\cdot) \rangle$ at points in the set

$$S := \{W' = (a', c', V') \,:\, a' = a_0, c' = c_0, \sqrt{\sum_{j \in [m]} \frac{1}{m} \left\| \frac{v'_j}{N(v_j)} - \frac{v_j}{N(v_j)} \right\|^2} \leq R_V\}$$

where $R_V$ is $\mathcal{O}(\gamma)$. We restrict $R_V$ as we cannot hope to compete against arbitarily large good reference parameter $V'$. We note that this radius is not as restrictive as it may appear since the term $\sqrt{\sum_{j \in [m]} \frac{1}{m} \left\| \frac{v'_j}{N(v_j)} - \frac{v_j}{N(v_j)} \right\|^2}$ is being normalized by $\sqrt{m}$. In particular, we are essentially competing with the optimal "effective" inner layer $V/N(V)$ that is within an $\mathcal{O}(\gamma\sqrt{m})$ ball centered at initialization. We now present the full assumption in detail.

**Assumption 1.2** (definition of $\gamma$). *For $j \in m$, let $(a_j, v_j) \sim \frac{\mathcal{N}_m}{\sqrt{m}} \times \frac{\mathcal{N}_{m \times d}}{\sqrt{d}}$ i.i.d. In addition, let $c = \overrightarrow{\mathbf{1}} \in \mathbb{R}^m$. There exists an $m' \geq 1$ such that if $m \geq m'$, we can find $m$ tuples $(\overrightarrow{a_j}, \overrightarrow{c_j}, \overrightarrow{v_j}) \in \mathbb{R} \times \mathbb{R} \times \mathbb{R}^d$ and constants $\gamma_a, \gamma_c \geq 0$ such that*

$$\left\| \overrightarrow{a} \right\|, \left\| \overrightarrow{C} \right\| \leq 1, \qquad \text{(norm condition)},$$

$$\left\langle \overrightarrow{a}, \partial_a f(x; W_0) \right\rangle \geq \gamma_a, \quad \left\langle \overrightarrow{C}, \partial_c f(x; W_0) \right\rangle \geq \gamma_c,$$

$$\gamma = \gamma_a + \gamma_c > 0, \qquad \text{(total margin is positive)},$$

$$\Delta_V := \sqrt{\sum_{j \in [m]} \frac{1}{m} \left\| \frac{\overrightarrow{v_j}}{N(v_j)} - \frac{v_j}{N(v_j)} \right\|^2} \leq \frac{\gamma}{32}, \quad \text{(max distance from initialization allowed)}.$$

We now calculate $\gamma$ on the following dyadic dataset.

**Definition 1.1** (Dyadic Dataset). *Let $e_1, \dots, e_d \in \mathbb{R}^d$ denote the standard basis vectors. Define examples $(x_i, y_i)_{i \in [2d]}$ as follows:*

$$(x_j, y_j) = (2^{-j} e_j, +1), \quad j \in \{1, \dots, d\}$$
$$(x_j, y_j) = (-2^{-j+d} e_{j-d}, -1), \quad j \in \{d+1, \dots, n\}$$

The purpose of this dataset is to highlight Batch Norm's ability to handle features of different order of magnitude.

**Proposition 1.1.** *Suppose we are given examples $(x_i, y_i)_{i \in [2d]}$ as defined in definition 1.1. Then the margin $\gamma_{linear}$ of the max margin linear predictor is*

$$\gamma_{linear} = \sqrt{\frac{3}{4^{d+1} - 4}}.$$

*Now consider the vector $u = (2, 4, \dots, 2^d)$ and define the unit vector $\overrightarrow{u} := \frac{u}{\|u\|}$. Taking $\overrightarrow{C} = 0$, $\overrightarrow{V} = V_0$, and $\overrightarrow{a} \in \mathbb{R}^m$ such that $\overrightarrow{a_j} = \text{sgn}(\overrightarrow{u}^\mathsf{T} v) \mathbf{1}_{\|v_{0_j}\| \in [\frac{1}{2}, 4]}$, we have that assumption 1.2 holds for $\gamma \geq \frac{\sqrt{d}}{40} \gamma_{linear}$.*

## 2 FORMAL STATEMENTS AND PROOF SKETCHES

We open the section with a theorem establishing low test error for SGD. In addition, for a network of width $m$, we show that at the time we reach low test error, the non-normalized parameters can move away from initialization by $O(m^{1/6})$ while the normalized parameters remain trapped in a $O(1)$ ball. We then present a similar theorem establishing low test error for GD. Unlike the SGD proof which converts a training error guarantee into a test error guarantee via a martingale concentration inequality, the GD proof requires a careful calculation of the Rademacher complexity of our function class.

**Theorem 2.1** (Stochastic Gradient Descent Guarantees). *Suppose weights $W = (a, c, V)$ where $(a, V) \sim \frac{\mathcal{N}_m}{\sqrt{m}} \times \frac{\mathcal{N}_{m \times d}}{\sqrt{d}}$, $c = \overrightarrow{\mathbf{1}} \in \mathbb{R}^m$ are given and assumption 1.2 holds, with corresponding $(\overrightarrow{a}, \overrightarrow{C}, \overrightarrow{V}) \in \mathbb{R}^m \times \mathbb{R}^m \times \mathbb{R}^{m \times d}$ and $\gamma = \gamma_a + \gamma_c$ given.*

*Suppose width $m$ satisfies*

$$m \in \left[ \frac{2^{29} \epsilon_N^3 \ln^{3/2}(\frac{5nt}{\delta})}{\gamma^2}, \delta \exp(\frac{d}{8}) \right]$$

*and we set the learning rate $\eta$ to be*

$$\eta = \frac{1}{28\epsilon_N \gamma^{2/3} m^{1/3}}.$$

*With probability at least $1 - 20\delta$,*

1. *the probability of misclassifying a point is*

$$\Pr(\text{sgn}(f(x; W)) \neq y) \leq \frac{2^{15}\epsilon_N^4}{\gamma^2 t} + \frac{4\ln(1/\delta)}{t}; \tag{2}$$

2. *the maximum distance the parameters can travel from initialization is*

$$\max_{s<t}\|U_s - U_0\| = \max_{s<t}\big\|(a_s, c_s) - (a_0, c_0)\big\| \leq \frac{\gamma^{1/3} m^{1/6}}{27\epsilon_N},$$

$$\max_{s<t}\|V_s - V_0\| \leq \frac{1}{2\epsilon_N}.$$

Here we provide a proof sketch for theorem 2.1.

Given scaling constant $r = O(\gamma^{1/3} m^{1/6})$, we first construct reference parameters

$$\overline{U} = (\overline{a}, \overline{C}) = r\overrightarrow{U} + U_0, \quad \overline{V} = V_0. \tag{3}$$

The reference parameters serve as good directions for weights of the network to travel toward. $\overline{U}$ will guide the iterates $\{W_s\}_{s<t}$ generated by SGD to a good solution.

1. We can use a perceptron style argument to control the loss derivatives

$$\sum_{s<t}|\ell'(W_s)| \leq \frac{\|U_\tau - U_0\|}{\gamma\sqrt{m}}, \tag{4}$$

   provided the iterates stay within a manifold $\Gamma := \{W' = (U, V) \mid \|U' - U\| < R, \|V' - V_0\| < R_V\}$ where radii $R, R_V > 0$ are determined a priori. One can then use the fact that asymptotically $\ell' \sim \ell$ and obtain a loss bound. Therefore, the remaining parts of the proof will establish that for carefully chosen radii $R, R_V$, the iterates remain trapped in $\Gamma$.

2. In order to trap the iterates $W_s$ in the manifold $\Gamma$, we assume contradictorily that there exists a first time $\tau \leq t$ where we exit the manifold $\Gamma$. To be more precise, letting rescaling constant $\alpha = \frac{R^2}{R_V^2}$, we consider the first time $\tau$ such that

$$\sqrt{\|U_\tau - U_0\|^2 + \alpha\|V_\tau - V_0\|^2} \geq R. \tag{5}$$

   While this exit condition is pessimistic ($W$ satisfying the exit condition implies $W \in \Gamma$ but the reverse statement is not necessarily true), checking one exit condition as opposed to two simplifies analysis. $\Gamma$ is similar to the manifold that was used to trap SGD iterates of shallow networks *without normalization* (Telgarsky, 2022). There, a simpler manifold was used $\{W' \mid \|W' - W\| \leq R\}$. The simpler manifold cannot be used here as a tighter control on $\|V_s - V_0\|$ is necessary. To see why, one can observe that small displacement in $V$ results in large changes in the reciprocal of the normalization factor $\frac{1}{N(v_j)}$. At first glance, it may seem rather strange normalization factors present a problem as they appear in conjunction with $\langle v, x \rangle$ in the network and thus the quantity $\frac{\langle v, x \rangle}{N(v)}$ is well behaved. Upon a closer inspection, one learns that the problem lies with the magnitude of the gradients. Consider the partial derivative of the predictions with respect to the inner layer $\partial_V p_i(W_s)$,

$$\sum \frac{a_j c_j o_j|_{\overline{x_i}}}{N(v_j)}\left(I - \frac{\Sigma v_{sj} v_{sj}^T}{N(v_{sj})^2}\right)e_j \overline{x_i}^\top.$$

Treating $\left( I - \frac{\Sigma v_j v_j^T}{N(v_j)^2} \right)$ as a bounded linear operator (which we show in our analysis) there is an extra reciprocal factor which cause a naive extension of (Telgarsky, 2022) to fail without careful control of $\|V_s - V_0\|$. Specifically, we force that it is at most $O(1)$ while still allowing the outer and affine layer $U = (a, C)$ to move $O(\gamma^{1/3} m^{1/6})$.

3. Now since for time $s < \tau$, we have $W_s \in \Gamma$, we will show that that the rescaled squared Euclidean potential satisfies

$$\Phi_D(\tau) = \left\| U_\tau - \overline{U} \right\|^2 - \left\| U_0 - \overline{U} \right\|^2 + \alpha \left( \left\| V_\tau - \overline{V} \right\|^2 - \left\| V_0 - \overline{V} \right\|^2 \right)$$

$$= \left\| U_\tau - \overline{U} \right\|^2 - \left\| U_0 - \overline{U} \right\|^2 + \alpha \| V_\tau - V_0 \|^2 \leq O(1).$$

This will be used to show that $\sqrt{\| U_\tau - U_0 \|^2 + \alpha \| V_\tau - V_0 \|^2} < R$ which is a contradiction.

4. Now that we have trapped all the iterates $W_s$ in $\Gamma$ we obtain a loss bound as detailed above. We then convert a training error bound into a test loss bound using a martingale concentration inequality.

We now provide a similar guarantee for GD.

**Theorem 2.2** (Gradient Descent Guarantees). *Suppose weights $W = (a, c, V)$ where $(a, V) \sim \frac{\mathcal{N}_m}{\sqrt{m}} \times \frac{\mathcal{N}_{m \times d}}{\sqrt{d}}$, $c = \overrightarrow{\mathbf{1}} \in \mathbb{R}^m$ are given and assumption 1.2 holds, with corresponding $(\overrightarrow{a}, \overrightarrow{C}, \overrightarrow{V}) \in \mathbb{R}^m \times \mathbb{R}^m \times \mathbb{R}^{m \times d}$ and $\gamma = \gamma_a + \gamma_c$ given.*

*Suppose width $m$ satisfies*

$$m \in \left[ \frac{2^{29} \epsilon_N^3 \ln^{3/2}(\frac{5nt}{\delta})}{\gamma^2}, \delta \exp(\frac{d}{8}) \right]$$

*and we set the learning rate $\eta$ to be*

$$\eta = \frac{1}{28 \epsilon_N \gamma^{2/3} m^{1/3}}.$$

*With probability at least $1 - 20\delta$,*

1. *the probability of misclassifying a point is*

$$\Pr(\text{sgn}(f(x; W)) \neq y) \leq \frac{2^{17} m^{1/3} \epsilon_N^4}{\gamma^{1/3} t \sqrt{n}} + \frac{4 \ln(1/\delta)}{t}; \tag{6}$$

2. *the maximum distance the parameters can travel from initialization is*

$$\max_{s < t} \| U_s - U_0 \| = \max_{s < t} \| (a_s, c_s) - (a_0, c_0) \| \leq 8,$$

$$\max_{s < t} \| V_s - V_0 \| \leq \frac{1}{2 \epsilon_N}.$$

The proof strategy for obtaining results for GD is relatively similar to that of SGD and mainly differs on converting training error guarantees to test error guarantees. The GD proof requires a careful calculation of the Rademacher complexity of our function class.

## 3 CONCLUSION AND OPEN PROBLEMS

**Feature learning.** The present analysis *requires* a displacement condition; can it be extended to a larger displacement radius, allowing feature learning regimes? Does batch norm and the rapid convergence in displacement (as above) have consequences on feature learning?

**Other architectures.** Can the proof techniques in the present work be extended to multi-layer cases, inhomogeneous cases, and other normalization types?

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

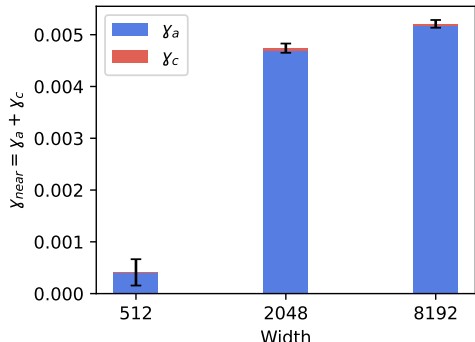

Figure 1: $\gamma$ estimates on the CIFAR-10 dataset. $\gamma_a$ contributes most of the value to $\gamma$ as $\gamma_c$ is relatively tiny.

## A  EXPERIMENTS

Here we provide details about the experiment setup and additional experiments.

### A.1  ESTIMATION OF $\gamma_a$ AND $\gamma_c$ ON CIFAR-10

In Figure 1 we provide empirical estimates for $\gamma$ on the subset of CIFAR-10 corresponding to the cats vs. dogs binary classification task, by estimating $\gamma_a$ and $\gamma_c$ via reference networks. See the subsection below on reference networks for more details. We try a few different widths and observe that larger widths tend to be better for achieving the best lower bound for $\gamma$. Error bars plotted are 2 standard errors, estimated with 5 random seeds.

### A.2  METHODS

**Dataset and Setup.** Experiments were conducted on the CIFAR-10 dataset (Krizhevsky et al., 2009), which uses the MIT license, and was downloaded via pytorch. To match the binary classification setting, we use only the data corresponding to the cat and dog classes, setting $y_i = -1$ for cat, and $y_i = 1$ for dog. To match the assumption used in the proofs, the data was normalized such that all the data points $x_i$ satisfied $\|x_i\| \leq 1$ .

We use a random subset of $n = 1000$ data points for the training set unless otherwise mentioned, rather than the full set of approximately $n = 12000$ points. This is due to the setting being batch GD, where $n = 12000$ takes a very long time per update, and requires larger networks and more iterations to achieve a positive margin. Since the primary goal of these experiments is to investigate values in practice versus theoretical bounds and compare BN and non-BN networks, rather than achieving good testing accuracy, we consider this a worthwhile tradeoff for the computational speedup.

The network architecture used is as stated in the main body of the paper, i.e. a two-layer ReLU net without biases, with BN unless otherwise stated. Similarly, initial parameters are sampled from the distributions given in the paper, whose scales deviate slightly from pytorch defaults. The networks are trained using batch GD on the logistic loss. Note that we flatten the spatial dimensions and RGB input channels into a single $32(32)(3) = 3072$ dimensional input vector.

**Hyperparameters.** For Figure 1, the learning rate ranges from 0.01 to 0.1, chosen to be as large as possible without causing significant instability or divergence in the optimization, depending on the network width. The networks are trained for as long as possible without being prohibitively long, in order to maximize the margin. The number of iterations does differ for networks of different widths, ranging from $t = 10000$ to $t = 30000$, but we observe by extrapolation that the margin reached at the end of training seems to be at least 0.75x of the expected final margin, regardless of where $t$ falls within $[10000, 30000]$.

**Compute and Resources.** The experiments required roughly 1 hour of compute on a V100, leveraging cloud computing resources from Google Colab.

**Reference Networks for Figure 1.** The explanation behind the method generating the $\gamma$ plot is somewhat involved. The goal of the experiments is to separately generate lower bound estimates for $\gamma_a$ and $\gamma_c$. To derive the process, note the inequalities in assumption 1.2 for $\gamma_a$ and $\gamma_c$. By instantiating arbitrary $(\overrightarrow{a_j}, \overrightarrow{c_j}, \overrightarrow{v_j})$ according to the norm constraints, we can set $\gamma_a$ and $\gamma_c$ equal to the left hand sides, yielding lower bounds for the best possible $\gamma_a$ and $\gamma_c$ (better instantiations would yield better $\gamma_a$ and $\gamma_c$).

Then, we choose to instantiate $(\overrightarrow{a_j}, \overrightarrow{c_j}, \overrightarrow{v_j})$ by training "reference networks" and extracting their parameters. Though it is not immediate, one can observe that the LHS for $\gamma_a$ corresponds to the unnormalized margin of a reference *a-network* divided by $\sqrt{m}$, where a reference *a-network* is a BN net with $a$ normalized such that $\|a\| \le 1$, $c$ frozen to its initialization value (hence $c = \overrightarrow{\mathbf{1}}$), and $V$ is constrained according to the displacement condition. Similarly, the LHS for $\gamma_c$ corresponds to a function of a reference *c-network*, which is a BN net with $a$ frozen to its initialization value, $c$ normalized such that $\|c\| \le 1$, and $V$ is constrained according to the displacement condition. Note that we may use arbitrary $(\overrightarrow{a_j}, \overrightarrow{c_j}, \overrightarrow{v_j})$, and thus we may generate these parameters any way we wish (not necessarily using batch GD). In this case, we use Adam (Kingma & Ba, 2014) instead of gradient descent to optimize these networks, and normalize them accordingly to find our lower bounds.

We remark that, since we may arbitrarily select $(\overrightarrow{a_j}, \overrightarrow{c_j}, \overrightarrow{v_j})$, we only use the above scheme to generate the parameters because it seems to be effective. We could also use optimization methods other than Adam/GD, different initialization scalings, or other setting modifications, provided that the process results in $(\overrightarrow{a_j}, \overrightarrow{c_j}, \overrightarrow{v_j})$ values that yield a good bound.

Lastly, constraining $V$ displacement is somewhat tricky. The allowed displacement depends on $\gamma = \gamma_a + \gamma_c$, so we need to estimate these terms first, but to do so according to our method, we need to generate (constrained-$V$) reference nets, so it is circular. To address this, we first get a lower bound on $\gamma$ by training reference networks with frozen $V$. Then we train reference networks allowing $V$ to move up to the amount prescribed by the displacement condition. However, in the given setup, we find that $V$ is allowed to move so little from initialization that allowing $V$ to move according to the displacement condition generates nearly equivalent $\gamma$ estimates as freezing $V$.

# B  DATA SEMI-NORM $\|\cdot\|_\Sigma$ AND NORMALIZATION FACTORS

This section develop tools to prevent normalization factors from becoming excessively small which is a core difficulty in the SGD/GF proofs.

We first derive useful relations between the data semi-norm $\|\cdot\|_\Sigma$, $\ell_2$ norm $\|\cdot\|$, and the normalization function $N(\cdot)$.

**Lemma B.1.**      *1. For any $v \in \mathbb{R}^d$, we have*
$$\|v\|_\Sigma \le \|v\| \quad \text{and} \quad \|\Sigma v\| \le \|v\|_\Sigma \le N(v).$$

   *2. Let $u, v \in \mathbb{R}^d$. Then we have*
$$\left| N(u) - N(v) \right| \le \|u - v\|_\Sigma$$

*Proof.*       1. We first show $\|v\|_\Sigma \le \|v\|$:
$$\|v\|_\Sigma^2 = \sum_{i \in [n]} \frac{\langle v, \overline{x_i} \rangle^2}{n} \le \sum_{i \in [n]} \frac{\|v\|^2}{n} = \|v\|^2. \tag{7}$$

   Taking the square root of both sides gives the desired claim.

   Now we show the sandwich inequality $\|\Sigma v\| \le \|v\|_\Sigma \le N(v)$. By definition of the normalization factor $N(\cdot)$, we have
$$N(v) = \sqrt{\|v\|_\Sigma^2 + \epsilon} \ge \|v\|_\Sigma.$$

   Since $\|v\|_\Sigma \le \|v\|$ by eq. (7),
$$\|\Sigma v\|^2 = \langle \Sigma v, \Sigma v \rangle = \langle \Sigma v, v \rangle_\Sigma \le \|\Sigma v\|_\Sigma \|v\|_\Sigma \le \|\Sigma v\| \|v\|_\Sigma.$$

   Dividing both sides by $\|\Sigma v\|$ gives us the desired claim.

2. Suppose $a, b \in \mathbb{R}$ such that $b \geq a \geq 0$. Consider the auxillary function

$$h(\epsilon) = \sqrt{b + \epsilon} - \sqrt{a + \epsilon}$$

Since $b + \epsilon \geq a + \epsilon$

$$h'(\epsilon) = \frac{1}{2}(\frac{1}{\sqrt{b + \epsilon}} - \frac{1}{\sqrt{a + \epsilon}}) \leq 0.$$

Therefore, for any $\epsilon \geq 0$, we have

$$h(\epsilon) \leq h(0).$$

Without loss of generality, assume $\|u\|_\Sigma \geq \|v\|_\Sigma$. Then

$$N(u) - N(v) = \sqrt{\|u\|_\Sigma^2 + \epsilon} - \sqrt{\|v\|_\Sigma^2 + \epsilon} \leq \|u\|_\Sigma - \|v\|_\Sigma \leq \|u - v\|_\Sigma.$$

Nonnegativity of the semi-norm $\|\cdot\|_\Sigma$ ensures

$$N(v) - N(u) \leq 0 \leq \|u - v\|_\Sigma.$$

$\square$

The following lemma argues that if for any given time $s$, the inner layer $V_s$ has not moved much (i.e. $\|V_s - V_0\| \leq O(1)$), then one can establish additive and multiplicative bounds for the reciprocal of the normalization factors $1/N(v_{sj})$. The actual lemma is a bit more general and applies for any $V' : \|V' - V\| \leq O(1)$. To simplify notation, we shall write $N(v_j)$ and $N(v'_j)$ as $N_j$ and $N'_j$ respectively.

**Lemma B.2.** *Let $V \in \mathbb{R}^{m \times d}$ and $v_j$ denote the rows of $V$. In addition, suppose $\epsilon_V, \epsilon_N > 0$ such that*

$$\|V^\top\|_{2,\infty} \leq \epsilon_V, \qquad\qquad \max_j \frac{1}{N_j} \leq \epsilon_N.$$

*For any $c < 1$, set*

$$\beta_V = \frac{c}{\epsilon_N}, \quad \rho_N = \frac{1}{1 - c}, \quad \beta_N = \rho_N \epsilon_N^2, \quad \beta_\Pi = \max(2, 2(\epsilon_V + \beta_V)\rho_N \epsilon_N), \quad \epsilon_{ratio} = \epsilon_V \epsilon_N.$$

*Then for any $j \in [m]$ and any $V' : \|V - V'\| \leq \beta_V$,*

1. *we have the following multiplicative and additive bound for the reciprocal of the normalization factors:*

$$\frac{1}{N'_j} \leq \rho_N \frac{1}{N_j} \quad and \quad \left|\frac{1}{N'_j} - \frac{1}{N_j}\right| \leq \beta_N \|v'_j - v_j\|.$$

2. *Define the operator $T_{v'_j} := I - \frac{\Sigma v'_j v'^\top_j}{N'^2_j}$ such that $T_{v'_j} : \mathbb{R}^d \to \mathbb{R}^d$. Then $T_{v'_j}$ is bounded:*

$$\left\|(I - \frac{\Sigma v'_j v'^\top_j}{N'^2_j})x\right\| \leq \beta_\Pi.$$

3. *We additionally have that $\left|\left\langle \frac{v_j}{N_j}, x \right\rangle\right| \leq \epsilon_{ratio}$ for all $i \in [n]$.*

*Proof.*      1. We first show the multiplicative bound $\frac{1}{N'_j} \leq \rho_N \frac{1}{N_j}$.

Since $\|V' - V\| \leq \beta_V = \frac{c}{\epsilon_N}$, for any $j \in [m]$,

$$\|v'_j - v_j\| \leq \|V' - V\| \leq \frac{c}{\epsilon_N} \leq c \min_k N(v_k) \leq cN_j. \tag{8}$$

Thus,

$$\frac{1}{N'_j} = \frac{1}{N_j + N'_j - N_j} \leq \frac{1}{N_j - \left|N'_j - N_j\right|}.$$

By lemma B.1 and eq. (8), we have that

$$\frac{1}{N_j - \left|N'_j - N_j\right|} \leq \frac{1}{N_j - \left\|v'_j - v_j\right\|_\Sigma} \leq \frac{1}{N_j - \left\|v'_j - v_j\right\|} \leq \frac{1}{N_j - cN_j} = \frac{1}{1 - c}\frac{1}{N_j}.$$

Recalling that $\rho_N = \frac{1}{1-c}$, we get the desired multiplicative bound $\frac{1}{N'_j} \leq \rho_N \frac{1}{N_j}$.

We now show the additive bound $\left|\frac{1}{N'_j} - \frac{1}{N_j}\right| \leq \beta_N \left\|v'_j - v_j\right\|$. Since $\frac{1}{N'_j} \leq \rho_N \frac{1}{N_j}$ by lemma B.2.1, we get

$$\left|\frac{1}{N_j} - \frac{1}{N_j}\right| = \frac{\left|N'_j - N_j\right|}{N_j N'_j} \leq \rho_N \frac{\left|N'_j - N_j\right|}{N_j^2}.$$

Finally, since $\left|N' - N_j\right| \leq \left\|v'_j - v_j\right\|_\Sigma \leq \left\|v'_j - v_j\right\|$ by lemma B.1 and $\max_j \frac{1}{N(v_j)} \leq \epsilon_N$, we get

$$\rho_N \frac{\left|N'_j - N_j\right|}{N_j^2} \leq \rho_N \epsilon_N^2 \left\|v'_j - v_j\right\|. \tag{9}$$

2. We consider two cases.

- **Case 1:** Suppose we had

$$\left|\left\langle v'_j, x\right\rangle\right| \leq \left|\left\langle v'_j, x\right\rangle_\Sigma\right|. \tag{10}$$

  Then, by lemma B.1 and our case assumption, eq. (10), expanding the norm gives

$$\left\|(I - \frac{\Sigma v'_j v'^\top_j}{N_j'^2})x\right\| = \sqrt{\|x\|^2 + \frac{\left\|\Sigma v'_j\right\|^2 \left\langle v'_j, x\right\rangle^2}{N_j'^4} - \frac{2\left\langle v', x\right\rangle_\Sigma \left\langle v', x\right\rangle}{N_j'^2}}$$

$$\leq \sqrt{\|x\|^2 + \frac{\left\|\Sigma v'_j\right\|^2 \left\langle v'_j, x\right\rangle_\Sigma^2}{N_j'^4} + \frac{2\left\langle v', x\right\rangle_\Sigma^2}{N_j'^2}} \quad \text{case assumption: eq. (10)}$$

$$\leq \sqrt{\|x\|^2 + \frac{\left\langle v'_j, x\right\rangle_\Sigma^2}{N_j'^2} + \frac{2\|v'\|_\Sigma^2 \|x\|_\Sigma^2}{N_j'^2}} \quad \text{lemma B.1.2}$$

$$\leq \sqrt{\|x\|^2 + \frac{\left\|v'_j\right\|_\Sigma^2 \|x\|_\Sigma^2}{N_j'^2} + \frac{2\|v\|_\Sigma^2 \|x\|_\Sigma^2}{N_j'^2}}$$

$$\leq \sqrt{\|x\|^2 + 3} \quad \text{lemma B.1.2}$$

$$\leq 2.$$

- **Case 2:** Suppose instead

$$\left|\left\langle v'_j, x\right\rangle\right| > \left|\left\langle v'_j, x\right\rangle_\Sigma\right|. \tag{11}$$

Since $\frac{1}{N_j'} \le \rho_N \frac{1}{N_j}$ by lemma B.2.1 and $\max_j \frac{1}{N_j} \le \epsilon_N$, we get

$$\frac{\left\| v_j' \right\|}{N_j'} \le \left( \|v_j\| + \left\| v_j' - v_j \right\| \right) \rho_N \epsilon_N \le (\epsilon_V + \beta_V) \rho_N \epsilon_N. \tag{12}$$

Expanding the norm and using lemma B.1, eqs. (11) and (12),

$$\left\| (I - \frac{\Sigma v_j' v_j'^{\mathsf{T}}}{N_j'^2}) x \right\| = \sqrt{ \|x\|^2 + \frac{\left\| \Sigma v_j' \right\|^2 \left\langle v_j', x \right\rangle^2}{N_j'^4} - 2 \frac{\left\langle v_j', x \right\rangle \left\langle v_j', x \right\rangle_\Sigma}{N_j'^2} }$$

$$\le \sqrt{ 1 + \frac{\left\langle v_j', x \right\rangle^2}{N_j'^2} + 2 \frac{\left| \left\langle v_j', x \right\rangle \right|}{N_j'} } \qquad \text{lemma B.1.2 and eq. (11)}$$

$$= 1 + \frac{\left| \left\langle v_j', x \right\rangle \right|}{N_j'}$$

$$\le 1 + \frac{\left\| v_j' \right\|}{N_j'}$$

$$\le 1 + (\epsilon_V + \beta_V) \rho_N \epsilon_N \qquad \text{eq. (12)}$$

$$\le \max(2, 2(\epsilon_V + \beta_V) \rho_N \epsilon_N).$$

3. Since $\|V^{\mathsf{T}}\|_{2,\infty} \le \epsilon_V$ and $\max_j \frac{1}{N_j} \le \epsilon_N$,

$$\left| \left\langle \frac{v_j}{N_j}, x \right\rangle \right| \le \frac{\|v_j\|}{N_j} \le \epsilon_V \epsilon_N. \tag{13}$$

$\square$

## C  GAUSSIAN CONCENTRATION

The following lemmas will control terms at initialization.

**Lemma C.1.** *Suppose we are given weights* $W = (a, c, V)$ *where* $(a, V) \sim \frac{\mathcal{N}_m}{\sqrt{m}} \times \frac{\mathcal{N}_{m \times d}}{\sqrt{d}}$ *and* $c = \overrightarrow{1} \in \mathbb{R}^m$. *Then, with probability at least* $1 - 4\delta$,

$$\max_{i \in [n]} \left| p_i(W) \right| = \max_{i \in [n]} \left| \sum_{j \in [m]} a_j \sigma(c_j \left\langle \frac{v_j}{N_j}, \overline{x_i} \right\rangle) \right| \le 4\epsilon_N \ln(\frac{n}{\delta}). \tag{14}$$

*Proof.* Fix index $i \in [n]$. We first massage the term $\left| p_i(W) \right|$ into a form where we can apply the uniform bound $\frac{1}{N_j} \le \epsilon_N$.

Let $e_j \in \mathbb{R}^m$ be the standard basis vector and $T_V$ denote the rotation matrix such that

$$T_V \begin{pmatrix} \sigma(c_1 \left\langle \frac{v_1}{N_1}, \overline{x_i} \right\rangle) \\ \vdots \\ \sigma(c_m \left\langle \frac{v_m}{N_m}, \overline{x_i} \right\rangle) \end{pmatrix} = \left\| \begin{pmatrix} \sigma(c_1 \left\langle \frac{v_1}{N_1}, \overline{x_i} \right\rangle) \\ \vdots \\ \sigma(c_m \left\langle \frac{v_m}{N_m}, \overline{x_i} \right\rangle) \end{pmatrix} \right\| e_1.$$

Define $\mu$ and $\mathcal{V}$ as the marginal distribution and random variable corresponding to $V$ respectively. Fix $\beta \geq 0$. By the law of total probability

$$\Pr(\left|\sum_{j \in [m]} a_j \sigma(c_j \left\langle \frac{v_j}{N_j}, \overline{x_i} \right\rangle)\right| > \beta) = \int \Pr(\left|\sum_{j \in [m]} a_j \sigma(c_j \left\langle \frac{v_j}{N_j}, \overline{x_i} \right\rangle)\right| > \beta \mid \mathcal{V} = V) \, d\mu(V)$$

$$= \int \Pr(\left|\left\langle a, \begin{pmatrix} \sigma(c_1 \left\langle \frac{v_1}{N_1}, \overline{x_i} \right\rangle) \\ \vdots \\ \sigma(c_m \left\langle \frac{v_m}{N_m}, \overline{x_i} \right\rangle) \end{pmatrix} \right\rangle\right| > \beta \mid \mathcal{V} = V) \, d\mu(V).$$

Let $g \sim \mathcal{N}(0, 1)$. By rotational invariance of gaussians, the tail probability can be rewritten as follows

$$\int \Pr(\left|\left\langle T_V^\intercal a, \begin{pmatrix} \sigma(c_1 \left\langle \frac{v_1}{N_1}, \overline{x_i} \right\rangle) \\ \vdots \\ \sigma(c_m \left\langle \frac{v_m}{N_m}, \overline{x_i} \right\rangle) \end{pmatrix} \right\rangle\right| \geq \beta \mid \mathcal{V} = V) \, d\mu(V)$$

$$= \int \Pr(|g| \left\|\begin{pmatrix} \sigma(c_1 \left\langle \frac{v_1}{N_1}, \overline{x_i} \right\rangle) \\ \vdots \\ \sigma(c_m \left\langle \frac{v_m}{N_m}, \overline{x_i} \right\rangle) \end{pmatrix}\right\| \geq \beta \mid \mathcal{V} = V) \, d\mu(V).$$

We have shown the follow random variables are equal in distribution.

$$\left|\sum_{j \in [m]} a_j \sigma(c_j \left\langle \frac{v_j}{N(v_j)}, \overline{x_i} \right\rangle)\right| \overset{d}{\sim} |g| \left\|\begin{pmatrix} \sigma(\left\langle c_1 \frac{v_1}{N(v_1)}, \overline{x_i} \right\rangle) \\ \vdots \\ \sigma(\left\langle c_m \frac{v_m}{N(v_m)}, \overline{x_i} \right\rangle) \end{pmatrix}\right\|. \tag{15}$$

Now we can apply the uniform bound $\frac{1}{N_j} \leq \epsilon_N$ to get that

$$\Pr(|g| \left\|\begin{pmatrix} \sigma(c_1 \left\langle \frac{v_1}{N_1}, \overline{x_i} \right\rangle) \\ \vdots \\ \sigma(c_m \left\langle \frac{v_m}{N_m}, \overline{x_i} \right\rangle) \end{pmatrix}\right\| > \beta) \leq \Pr(\epsilon_N |g| \left\|\begin{pmatrix} \sigma(c_1 \left\langle v_1, \overline{x_i} \right\rangle) \\ \vdots \\ \sigma(c_m \left\langle v_m, \overline{x_i} \right\rangle) \end{pmatrix}\right\| > \beta).$$

Recalling that $c = \mathbf{1}$, we have altogther

$$\Pr\left(\left|\sum_{j \in [m]} a_j \sigma(c_j \left\langle \frac{v_j}{N_j}, \overline{x_i} \right\rangle)\right| > \beta\right) \leq \Pr(|g| \left\|\begin{pmatrix} \sigma(\left\langle \frac{v_1}{N_1}, \overline{x_i} \right\rangle) \\ \vdots \\ \sigma(\left\langle \frac{v_m}{N_m}, \overline{x_i} \right\rangle) \end{pmatrix}\right\| > \beta). \tag{16}$$

Using the same argument we used to derive eq. (15), one can show that

$$\epsilon_N \left|\sum_{j \in [m]} a_j \sigma(\left\langle v_j, \overline{x_i} \right\rangle)\right| \overset{d}{\sim} \epsilon_N |g| \left\|\begin{pmatrix} \sigma(\left\langle v_1, \overline{x_i} \right\rangle) \\ \vdots \\ \sigma(\left\langle v_m, \overline{x_i} \right\rangle) \end{pmatrix}\right\|. \tag{17}$$

By eq. (16) and eq. (17),

$$
\Pr\left( \left| \sum_{j\in[m]} a_j \sigma(c_j \left\langle \frac{v_j}{N_j}, \overline{x_i} \right\rangle) \right| \ge 4\epsilon_N \ln(1/\delta) \right) \le \Pr\left( \epsilon_N \left| \sum_{j\in[m]} a_j \sigma(\langle v_j, \overline{x_i} \rangle) \right| > 4\epsilon_N \ln(1/\delta) \right).
$$

By the proof of (Telgarsky, 2022, lemma A.2), we have that with probability $1 - 4\delta$,

$$
\Pr\left( \max_i \epsilon_N \left| \sum_{j\in[m]} a_j \sigma(\langle v_j, \overline{x_i} \rangle) \right| > 4\epsilon_N \ln(1/\delta) \right) \le 4\delta.
$$

Therefore, union bounding over $i \in [n]$, we have with probability at least $1 - 4\delta$,

$$
\max_i \left| \sum_{j\in[m]} a_j \sigma(c_j \left\langle \frac{v_j}{N_j}, \overline{x_i} \right\rangle) \right| \le 4\epsilon_N \ln(n/\delta).
$$

$\square$

The following lemma controls norms of gaussian vectors.

**Lemma C.2.** *Suppose* $(a, V) \sim \frac{\mathcal{N}_m}{\sqrt{m}} \times \frac{\mathcal{N}_{m \times d}}{\sqrt{d}}$. *With probability at least* $1 - \delta$, *we have that*

$$
\|a\| \le 1 + \sqrt{\frac{2\ln(\frac{1}{\delta})}{m}}.
$$

*Additionally, if* $m \le \delta \exp(\frac{d}{8})$,

- *with probability at least* $1 - \delta$, *we have*

$$
\left\| V^\intercal \right\|_{2,\infty} \le 1 + \sqrt{\frac{2\ln(\frac{1}{\delta})}{d}} \le 2.
$$

- *Suppose* $d \ge 2$. *With probability at least* $1 - \delta$,

$$
\max_{j\in m} \frac{1}{\|v_j\|} \le 5.
$$

*Proof.* We mostly repeat the proof of (Telgarsky, 2022, lemma B.3). Let $\widetilde{a} = a\sqrt{m}$ so that $\widetilde{a} \sim \mathcal{N}_m$. Since the function $\widetilde{a} \to \frac{\|\widetilde{a}\|}{\sqrt{m}}$ is $(1/\sqrt{m})$- lipschitz, by gaussian concentration (Wainwright, 2019, theorem 2.26), with probability at least $1 - \delta$

$$
\|a\| = \|\widetilde{a}\| / \sqrt{m}
$$

$$
\le \mathbb{E}\|\widetilde{a}\| / \sqrt{m} + \sqrt{\frac{2\ln(1/\delta)}{m}}
$$

$$
\le \sqrt{\mathbb{E}\|\widetilde{a}\|^2} / \sqrt{m} + \sqrt{\frac{2\ln(1/\delta)}{m}}
$$

$$
= 1 + \sqrt{\frac{2\ln(1/\delta)}{m}}.
$$

By the same argument, with probability at least $1 - \delta$, we have

$$
\|v_j\| \le 1 + \sqrt{\frac{2\ln(1/\delta)}{d}}.
$$

Union bounding $j \in [m]$ and recalling that $m \leq \delta \exp(\frac{d}{8})$, we have with probability at least $1 - \delta$

$$\max_{j \in [m]} \|v_j\| \leq 1 + \sqrt{\frac{2\ln(m/\delta)}{d}} \leq 1 + \sqrt{\frac{1}{4}} < 2.$$

Additionally, for any fixed $j \in [m]$, we have with probability at least $1 - \delta$ ((Wainwright, 2019, theorem 2.26))

$$\|v_j\| \geq \mathbb{E}[\|v_j\|] - \sqrt{\frac{2\ln(1/\delta)}{d}} = \sqrt{\frac{2}{d}} \frac{\Gamma(\frac{d+1}{2})}{\Gamma(\frac{d}{2})} - \sqrt{\frac{2\ln(1/\delta)}{d}}. \tag{18}$$

Let us recall Gautschi's inequality which states that for any $x \in \mathbb{R}$ and $s \in (0, 1)$,

$$x^{1-s} < \frac{\Gamma(x+1)}{\Gamma(x+s)} < (x+1)^{1-s}.$$

Setting $x = \frac{d}{2} - \frac{1}{2}$ and $s = \frac{1}{2}$, Gautschi's inequality asserts $\frac{\Gamma(\frac{d+1}{2})}{\Gamma(\frac{d}{2})} > \sqrt{\frac{d-1}{2}}$. Thus, eq. (18) can be simplified:

$$\|v_j\| > \sqrt{\frac{2}{d}}\sqrt{\frac{d-1}{d}} - \sqrt{\frac{2\ln(1/\delta)}{d}} = \sqrt{\frac{1}{2}} - \sqrt{\frac{2\ln(1/\delta)}{d}}.$$

Therefore, union bounding over $j \in [m]$,

$$\min_{j \in [m]} \|v_j\| \geq \sqrt{\frac{1}{2}} - \sqrt{\frac{2\ln(1/\delta)}{d}}$$

$$= \sqrt{\frac{1}{2}} - \sqrt{\frac{2\ln(m/\delta)}{d}}$$

$$\geq \sqrt{\frac{1}{2}} - \sqrt{\frac{1}{4}}$$

$$> \frac{1}{5}.$$

Rearranging the terms gives the desired claim.

$\square$

## D  ALIGNMENT

This section will establish that reference parameters constructed from $\overrightarrow{W} = (\overrightarrow{a}, \overrightarrow{C}, \overrightarrow{V})$, corresponding to assumption 1.2, is highly aligned with gradient features $\partial p_i(W')$ where $W'$ is constrained to some manifold $\Gamma$. Large alignment in turn will be used to guide SGD and GF iterates $W_s$ toward good directions in the proofs of theorem 2.1 and theorem 2.2.

The following lemma will control common terms found when trying to bound various alignment quantities in lemma D.3.

**Lemma D.1.** *Fix $V, \overline{V} \in \mathbb{R}^{m \times d}$ and let $(\beta_V, \beta_N, \beta_\Pi, \epsilon_{ratio})$ be constants corresponding to $V$ in the context of lemma B.2. Then for any $V'$ such that $\|V - V'\| \leq \beta_V$, we have*

1. *an uniform upper bound for the difference between per-row preactivations corresponding to $V, V'$*

$$\left| \left\langle \frac{v'_j}{N'_j}, \overline{x_i} \right\rangle - \left\langle \frac{v_j}{N_j}, \overline{x_i} \right\rangle \right| \leq (\rho_N \epsilon_N + \beta_N \epsilon_{ratio}) \|v'_j - v_j\|,$$

2. *an uniform upper bound for per-row preactivations corresponding to $V'$*

$$\left| \left\langle \frac{v'_j}{N'_j}, \overline{x_i} \right\rangle \right| \leq (\rho_N \epsilon_N + \beta_N \epsilon_{ratio}) \|v'_j - v_j\| + \epsilon_{ratio},$$

3. *an uniform upper bound for the difference between per-row preactivations corresponding to $V, \overline{V}$*

$$\left| \left\langle \frac{v'_j}{N'_j}, \overline{x_i} \right\rangle - \langle \overline{v_j}, \overline{x_i} \rangle \right| \le (\rho_N \epsilon_N + \beta_N \epsilon_{ratio}) \left\| v'_j - v_j \right\| + \left\| \frac{v_j}{N_j} - \overline{v_j} \right\|,$$

4. *an uniform upper bound for per-row preactivations corresponding to $\overline{V}$*

$$\left| \langle \overline{v_j}, \overline{x_i} \rangle \right| \le \left\| \overline{v_j} - \frac{v_j}{N_j} \right\| + \epsilon_{ratio},$$

5. *an uniform upper bound for the difference between per-row activations corresponding to $V, V'$*

$$\left| o'_j \big|_{\overline{x_i}} \left\langle \frac{v'_j}{N'_j}, \overline{x_i} \right\rangle - o_j \big|_{\overline{x_i}} \langle \overline{v_j}, \overline{x_i} \rangle \right| \le (\rho_N \epsilon_N + \beta_N \epsilon_{ratio}) \left\| v'_j - v_j \right\| + \left\| \frac{v_j}{N_j} - \overline{v_j} \right\| + \epsilon_{ratio}.$$

*Proof.*    1. Lemma B.2 establishes the following upper bounds:

$$\frac{1}{N'_j} \le \rho_N \frac{1}{N_j}, \qquad \left| \frac{1}{N'_j} - \frac{1}{N_j} \right| \le \beta_N \left\| v'_j - v_j \right\|, \qquad \left| \left\langle \frac{v_j}{N_j}, x \right\rangle \right| \le \epsilon_{ratio}$$

which in conjuction with the fact that $\max_{j \in [m]} \frac{1}{N_j} \le \epsilon_N$ grants

$$\left| \left\langle \frac{v'_j}{N'_j}, \overline{x_i} \right\rangle - \left\langle \frac{v_j}{N_j}, \overline{x_i} \right\rangle \right| \le \frac{\left\| v'_j - v_j \right\|}{N'_j} + \left| \frac{1}{N'_j} - \frac{1}{N_j} \right| \left| \left\langle \frac{v_j}{N_j}, \overline{x_i} \right\rangle \right|$$

$$\le \rho_N \frac{\left\| v'_j - v_j \right\|}{N_j} + \beta_N \left\| v'_j - v_j \right\| \epsilon_{ratio}$$

$$\le (\rho_N \epsilon_N + \beta_N \epsilon_{ratio}) \left\| v'_j - v_j \right\|. \tag{19}$$

2. By eq. (19) and since $\left| \left\langle \frac{v_j}{N_j}, x \right\rangle \right| \le \epsilon_{ratio}$ (lemma B.2), we have

$$\left| \left\langle \frac{v'_j}{N'_j}, \overline{x_i} \right\rangle \right| \le \left| \left\langle \frac{v'_j}{N'_j}, \overline{x_i} \right\rangle - \left\langle \frac{v_j}{N_j}, \overline{x_i} \right\rangle \right| + \left| \left\langle \frac{v_j}{N_j}, \overline{x_i} \right\rangle \right|$$

$$\le (\rho_N \epsilon_N + \beta_N \epsilon_{ratio}) \left\| v'_j - v_j \right\| + \epsilon_{ratio}. \tag{20}$$

3. Again by employing eq. (19),

$$\left| \left\langle \frac{v'_j}{N'_j}, \overline{x_i} \right\rangle - \langle \overline{v_j}, \overline{x_i} \rangle \right| \le \left| \left\langle \frac{v'_j}{N'_j}, \overline{x_i} \right\rangle - \left\langle \frac{v_j}{N_j}, \overline{x_i} \right\rangle \right| + \left| \left\langle \frac{v_j}{N_j}, \overline{x_i} \right\rangle - \langle \overline{v_j}, \overline{x_i} \rangle \right|$$

$$\le (\rho_N \epsilon_N + \beta_N \epsilon_{ratio}) \left\| v'_j - v_j \right\| + \left\| \frac{v_j}{N_j} - \overline{v_j} \right\|. \tag{21}$$

4. Since $\left| \left\langle \frac{v_j}{N_j}, x \right\rangle \right| \le \epsilon_{ratio}$ by lemma B.2,

$$\left| \langle \overline{v_j}, \overline{x_i} \rangle \right| \le \left| \langle \overline{v_j}, \overline{x_i} \rangle - \left\langle \frac{v_j}{N_j}, \overline{x_i} \right\rangle \right| + \left| \left\langle \frac{v_j}{N_j}, \overline{x_i} \right\rangle \right|$$

$$\le \left\| \overline{v_j} - \frac{v_j}{N_j} \right\| + \epsilon_{ratio}. \tag{22}$$

5. By eqs. (20) to (22), we immediately have that

$$\max\left(\left|\left\langle\frac{v_j'}{N_j'},\overline{x_i}\right\rangle\right|,\left|\langle\overline{v_j},\overline{x_i}\rangle\right|,\left|\left\langle\frac{v_j'}{N_j'},\overline{x_i}\right\rangle-\langle\overline{v_j},\overline{x_i}\rangle\right|\right)\leq(\rho_N\epsilon_N+\beta_N\epsilon_{\text{ratio}})\left\|v_j'-v_j\right\|+\left\|\frac{v_j}{N_j}-\overline{v_j}\right\|+\epsilon_{\text{ratio}}$$

which in turn controls

$$\left|o_j'\big|_{\overline{x_i}}\left\langle\frac{v_j'}{N_j'},\overline{x_i}\right\rangle-o_j\big|_{\overline{x_i}}\langle\overline{v_j},\overline{x_i}\rangle\right|=\begin{cases}\left|\left\langle\frac{v_j'}{N_j'},\overline{x_i}\right\rangle\right| & o_j'\big|_{\overline{x_i}}=1,o_j\big|_{\overline{x_i}}=0\\[2mm]\left|\langle\overline{v_j},\overline{x_i}\rangle\right| & o_j'\big|_{\overline{x_i}}=0,o_j\big|_{\overline{x_i}}=1\\[2mm]\left|\left\langle\frac{v_j'}{N_j'},\overline{x_i}\right\rangle-\langle\overline{v_j},\overline{x_i}\rangle\right| & o_j'\big|_{\overline{x_i}}=1,o_j\big|_{\overline{x_i}}=1\\[2mm]0 & o_j'\big|_{\overline{x_i}}=o_j\big|_{\overline{x_i}}=0\end{cases}$$

$$\leq(\rho_N\epsilon_N+\beta_N\epsilon_{\text{ratio}})\left\|v_j'-v_j\right\|+\left\|\frac{v_j}{N_j}-\overline{v_j}\right\|+\epsilon_{\text{ratio}}.$$

$\square$

The following technical lemma will be used to prove lemma D.3.

**Lemma D.2.** *Suppose weights $W=(a,c,V)$ where $(a,V)\sim\frac{\mathcal{N}_m}{\sqrt{m}}\times\frac{\mathcal{N}_{m\times d}}{\sqrt{d}}$, $c=\vec{1}\in\mathbb{R}^m$, and $\delta\in(0,1)$ are given. Consider radii $R,R_V$, rescaling constant $\alpha$, and width $m$ satisfying*

$$R\leq\frac{\gamma^{1/3}m^{1/6}}{27\epsilon_N},\qquad R_V=\beta_V=\frac{1}{2}\epsilon_N,\qquad\alpha=\frac{R^2}{\beta_V^2},\qquad m\leq\delta\exp(\frac{d}{2}).$$

*Denote $\Gamma:=\{W'\mid\|U'-U\|<R,\|V'-V\|<R_V\}$. Then, with probability at least $1-6\delta$, for any $W'\in\Gamma$,*

$$\left|\langle a,\partial_a p_i(W')\rangle\right|+\left|\langle C'-C,\partial_c p_i(W')\rangle\right|+\alpha\left|\langle V'-V,\partial_V p_i(W')\rangle\right|\leq R\frac{\gamma\sqrt{m}}{32}.$$

*Proof.* By lemma C.2, with probability at least $1-2\delta$, $\|V^\top\|_{2,\infty}\leq2$ and $\|a\|\leq2$. Given that $\beta_V=\frac{1}{2\epsilon_N}$, let $(\rho_N,\beta_N,\beta_\Pi,\epsilon_{\text{ratio}})$ be constants corresponding to $V$ in lemma B.2. In particular, we can consider the following instantiation

$$\rho_N=2,\qquad\beta_N=2\epsilon_N^2,\qquad\beta_\Pi=9\epsilon_N,\qquad\epsilon_{\text{ratio}}=2\epsilon_N.\qquad(23)$$

Now, let us start by bounding by $\left|\langle a,\partial_a p_i(W')\rangle\right|$. We first introduce terms $\sum_{j\in[m]}a_j\sigma(c_j\left\langle\frac{v_j'}{N_j'},\overline{x_i}\right\rangle)$ and $\sum_{j\in[m]}a_j\sigma(c_j\left\langle\frac{v_j}{N_j},\overline{x_i}\right\rangle)$ via add and subtract, and then we apply Cauchy-Schwarz to get

$$\left|\langle a,\partial_a p_i(W')\rangle\right|$$

$$=\left|\sum_{j\in[m]}a_j\sigma(c_j'\left\langle\frac{v_j'}{N_j'},\overline{x_i}\right\rangle)\right|$$

$$\leq\left|\sum_{j\in[m]}a_j\left(\sigma(c_j'\left\langle\frac{v_j'}{N_j'},\overline{x_i}\right\rangle)-\sigma(c_j\left\langle\frac{v_j'}{N_j'},\overline{x_i}\right\rangle)+\sigma(c_j\left\langle\frac{v_j'}{N_j'},\overline{x_i}\right\rangle)-\sigma(c_j\left\langle\frac{v_j}{N_j},\overline{x_i}\right\rangle)+\sigma(c_j\left\langle\frac{v_j}{N_j},\overline{x_i}\right\rangle)\right)\right|$$

$$\leq\sum_{j\in[m]}|a_j|\left|c_j'-c_j\right|\left|\left\langle\frac{v_j'}{N_j'},\overline{x_i}\right\rangle\right|+|a_j||c_j|\left|\left\langle\frac{v_j'}{N_j'},\overline{x_i}\right\rangle-\left\langle\frac{v_j}{N_j},\overline{x_i}\right\rangle\right|+\left|\sum_{j\in[m]}a_j\sigma(c_j\left\langle\frac{v_j}{N_j},\overline{x_i}\right\rangle)\right|.$$

$$(24)$$

Since $W' \in \Gamma$, we have that $\max_j \left\| v'_j - v_j \right\| \leq \left\| V' - V \right\| \leq \beta_V$. Therefore, we can invoke lemma D.1.2 and lemma D.1.1 bound to $\left| \left\langle \frac{v'_j}{N'_j}, \overline{x_i} \right\rangle \right|$ and $\left| \left\langle \frac{v'_j}{N'_j}, \overline{x_i} \right\rangle - \left\langle \frac{v_j}{N_j}, \overline{x_i} \right\rangle \right|$ respectively. Discarding an additional $4\delta$ failure probability, we apply lemma C.1 to get that $\left| \sum_{j \in [m]} a_j \sigma(c_j \left\langle \frac{v_j}{N_j}, \overline{x_i} \right\rangle) \right| \leq 4\epsilon_N \ln(\frac{n}{\delta})$ whereby the eq. (24) simplifies to

$$
\left| \langle a, \partial_a p_i(W') \rangle \right|
$$

$$
\leq \sum_{j \in [m]} \left| a_j \right| \left| c'_j - c_j \right| \left( (\rho_N \epsilon_N + \beta_N \epsilon_{\text{ratio}}) \left\| v'_j - v_j \right\| + \epsilon_{\text{ratio}} \right) + \sum_{j \in [m]} \left| a_j \right| (\rho_N \epsilon_N + \beta_N \epsilon_{\text{ratio}}) \left\| v'_j - v_j \right\| + 4\epsilon_N \ln(\frac{n}{\delta})
$$

$$
\leq \|a\| \|C' - C\| ((\rho_N \epsilon_N + \beta_N \epsilon_{\text{ratio}})\beta_V + \epsilon_{\text{ratio}}) + \|a\| (\rho_N \epsilon_N + \beta_N \epsilon_{\text{ratio}})\beta_V + 4\epsilon_N \ln(\frac{n}{\delta})
$$

$$
\leq 10\epsilon_N^2 R + 6\epsilon_N^2 + 4\epsilon_N \ln(n/\delta).
$$

We now tackle the second term $\left| \langle C' - C, \partial_c p_i(W') \rangle \right|$. Lemma D.1.2 establishes

$$
\left| \left\langle \frac{v'_j}{N'_j}, \overline{x_i} \right\rangle \right| \leq (\rho_N \epsilon_N + \beta_N \epsilon_{\text{ratio}}) \left\| v'_j - v_j \right\| + \epsilon_{\text{ratio}}.
$$

The preceding ienquality and multiple applications of Cauchy-Schwarz implies

$$
\left| \langle C' - C, \partial_c p_i(W_s) \rangle \right| = \left| \sum_{j \in [m]} (c'_j - c_j) \left[ a_j o'_j |_{\overline{x_i}} \left\langle \frac{v'_j}{N'_j}, \overline{x_i} \right\rangle \right] \right|
$$

$$
\leq \sum_{j \in [m]} \left| c'_j - c_j \right| (|a_j - a_j| + |a_j|) \left| \left\langle \frac{v'_j}{N'_j}, \overline{x_i} \right\rangle \right|
$$

$$
\leq \sum_{j \in [m]} \left| c'_j - c_j \right| (|a_j - a_j| + |a_j|) \left( (\rho_N \epsilon_N + \beta_N \epsilon_{\text{ratio}}) \left\| v'_j - v_j \right\| + \epsilon_{\text{ratio}} \right)
$$

$$
\leq \|C' - C\| \left( \|a' - a\| + 2 \right) \left( (\rho_N \epsilon_N + \beta_N \epsilon_{\text{ratio}}) \|V' - V\| + \epsilon_{\text{ratio}} \right)
$$

$$
\leq 5\epsilon_N^2 (R^2 + 2R).
$$

Finally, we control $\alpha \left| \langle V' - V, \partial_V p_i(W') \rangle \right|$. Since $\left\| (I - \frac{\Sigma v'_j v'^{\mathsf{T}}_j}{N'^2_j}) x \right\| \leq \beta_\Pi$ by lemma B.2, we have

$$
\alpha \left| \langle V' - V, \partial_V p_i(W_S) \rangle \right| = \alpha \left| \sum_{j \in [m]} \left\langle v'_j - v_j, \frac{a_j c'_j o'_j |_{\overline{x_i}}}{N'_j} \left[ \overline{x_i} - \frac{\Sigma v'_j \left\langle v'_j, \overline{x_i} \right\rangle}{N'^2_j} \right] \right\rangle \right|
$$

$$
\leq \alpha \sum_{j \in [m]} \left\| v'_j - v_j \right\| \left| \frac{a_j c'_j}{N'_j} \right| \left\| \left( I - \frac{v'_j v'^{T}_j}{N'^2_j} \right) \overline{x_i} \right\|
$$

$$
\leq \alpha \beta_\Pi \sum_{j \in [m]} \left\| v'_j - v_j \right\| \left| \frac{a_j c'_j}{N'_j} \right|
$$

$$
\leq \alpha \beta_\Pi \sum_{j \in [m]} \frac{\left\| v'_j - v_j \right\|}{N'_j} \left[ \left| a_j - a_j \right| \left| c'_j - c_j \right| + \left| c_j \right| \left| a_j - a_j \right| + \left| a_j \right| \left| c'_j - c_j \right| + \left| a_j \right| \left| c_j \right| \right].
$$

$$
\tag{25}
$$

Recalling that $c = \vec{\mathbf{1}}$ and applying the inequality $\frac{1}{N'_j} \leq \rho_N \epsilon_N$ (lemma B.2), Equation (25) simplifies to

$$
\begin{aligned}
\alpha \left| \left\langle V' - V, \partial_V p_i(W_S) \right\rangle \right| &\leq \alpha \beta_\Pi \sum_{j \in [m]} \frac{\left\| v'_j - v_j \right\|}{N'_j} \left[ \left| a_j - a_j \right| \left| c'_j - c_j \right| + \left| a_j - a_j \right| + \left| a_j \right| \left| c'_j - c_j \right| + \left| a_j \right| \right] \\
&\leq \alpha \beta_\Pi \sum_{j \in [m]} \left\| v'_j - v_j \right\| (\rho_N \epsilon_N) \left[ \left| a_j - a_j \right| \left| c'_j - c_j \right| + \left| a_j - a_j \right| + \left| a_j \right| \left| c'_j - c_j \right| \right] \\
&\leq \alpha \beta_\Pi \rho_N \epsilon_N \left\| V' - V \right\| \left( \left\| a' - a \right\| \left\| C' - C \right\| + \left\| a' - a \right\| + \left\| a \right\| \left\| C' - C \right\| \right) \\
&\leq 36 \epsilon_N^2 (R^4 + 3R^3).
\end{aligned}
$$

Putting everything together gives

$$
\begin{aligned}
\left| \left\langle a, \partial_a p_i(W') \right\rangle \right| &+ \left| \left\langle C' - C, \partial_c p_i(W') \right\rangle \right| + \alpha \left| \left\langle V' - V, \partial_V p_i(W') \right\rangle \right| \\
&\leq 36 \epsilon_N^3 R^4 + 108 \epsilon_N^3 R^3 + 5 \epsilon_N^2 R^2 + 20 \epsilon_N^2 R + 6 \epsilon_N^2 + 4 \epsilon_N \ln(n/\delta) \\
&\leq R \frac{\gamma \sqrt{m}}{32}.
\end{aligned}
$$

$\square$

The following lemma will control important alignment quantities. The first half will control alignment with gradient features at special points. The remaining half will control alignment with gradient features generated from $W' \in \Gamma$. Before stating the lemma, we introduce further notation. We shall overload the normalization function $N(\cdot)$ so that given $V \in \mathbb{R}^{m \times d}$, $N(V) = (N(v_1), \ldots, N(v_m))$. In additon, $\odot$ will denote element-wise multiplication.

**Lemma D.3.** *Suppose weights $W = (a, c, V)$ where $(a, V) \sim \frac{\mathcal{N}_m}{\sqrt{m}} \times \frac{\mathcal{N}_{m \times d}}{\sqrt{d}}$, $c = \vec{\mathbf{1}} \in \mathbb{R}^m$ are given and assumption 1.2 holds, with corresponding $(\vec{a}, \vec{C}, \vec{V}) \in \mathbb{R}^m \times \mathbb{R}^m \times \mathbb{R}^{m \times d}$ and $\gamma = \gamma_a + \gamma_c$ given.*

*In addition, consider radii $R, R_V, r$, rescaling constant $\alpha$, and width $m$ satisfying*

$$
R \in \left[ 8, \frac{\gamma^{1/3} m^{1/6}}{27 \epsilon_N} \right], \quad R_V \leq \beta_V = \frac{1}{2\epsilon_N}, \quad r = \frac{R}{8}, \quad \alpha = \frac{R^2}{\beta_V^2}, \quad m \in \left[ \frac{2^{29} \epsilon_N^3 \ln^{3/2}(5n/\delta)}{\gamma^2}, \delta \exp(\frac{d}{8}) \right].
$$

*Further, construct the following reference parameters*

$$
\overline{U} = (\overline{a}, \overline{C}) = r\vec{U} + U, \quad \overline{W^a} = (\overline{a}, N(\vec{V}) \odot C, \vec{V}), \quad \overline{W^C} = (a, N(\vec{V}) \odot \vec{C}, \vec{V}). \quad (26)
$$

*Denote $\Gamma := \{ W' \mid \left\| U' - U \right\| < R, \left\| V' - V \right\| < R_V \}$. Then, with probability at least $1 - 7\delta$, for any $W' \in \Gamma$ we have*

1. $\left\langle \vec{U}, \partial_U p_i(W') \right\rangle \geq \frac{3\gamma\sqrt{m}}{4}$,

2. $\left\langle \overline{U}, \partial_U p_i(W') \right\rangle - p_i(W') + \alpha \left\langle V - V', \partial_V p_i(W') \right\rangle \geq r(\frac{\gamma\sqrt{m}}{2})$.

*Proof.* By lemma C.2, with probability at least $1 - \delta$, $\left\| V^\top \right\|_{2,\infty} \leq 2$. Given that $\beta_V = \frac{1}{2\epsilon_N}$, let $(\rho_N, \beta_N, \beta_\Pi, \epsilon_{\text{ratio}})$ be constants corresponding to $V$ in lemma B.2. Without loss of generality, assume $\epsilon_N \geq 1$. Then, we can simplify the constants:

$$
\rho_N = 2, \qquad \beta_N = 2\epsilon_N^2, \qquad \beta_\Pi = 9\epsilon_N, \qquad \epsilon_{\text{ratio}} = 2\epsilon_N. \quad (27)
$$

By assumption 1.2 and definition of $\overline{W^a}$ and $\overline{W^c}$, we have

$$\left\langle \overrightarrow{a}, \partial_a p_i(\overline{W^a}) \right\rangle \geq \gamma_a \sqrt{m} \quad \text{and} \quad \left\langle \overrightarrow{C}, \partial_c p_i(\overline{W^C}) \right\rangle \geq \gamma_c \sqrt{m}. \tag{28}$$

Further, since $W' \in \Gamma$, we have

$$\max\left(\|a' - a\|, \|C' - C\|\right) \leq R \quad \text{and} \quad \|V' - V\| \leq \beta_V.$$

With these facts in hand, let us prove the two alignment inequalities.

1. We first note that

$$\left| \left\langle \overrightarrow{a}, \partial_a p_i(W_s) - \partial_a p_i(\overline{W^a}) \right\rangle \right|$$

$$\leq \sum_{j \in [m]} |\overrightarrow{a_j}| \left| \sigma\left(c'_j \left\langle \frac{v'_j}{N'_j}, \overline{x_i} \right\rangle\right) - \sigma\left(c_j \langle \overline{v_j}, \overline{x_i} \rangle\right) \right|$$

$$\leq \sum_{j \in [m]} |\overrightarrow{a_j}| \left( |c'_j - c_j| \left| \left\langle \frac{v'_j}{N'_j}, \overline{x_i} \right\rangle \right| + |c_j| \left| \left\langle \frac{v'_j}{N'_j}, \overline{x_i} \right\rangle - \langle \overline{v_j}, \overline{x_i} \rangle \right| \right).$$

Using lemma D.1.2 to get $\left| \left\langle \frac{v'_j}{N'_j}, \overline{x_i} \right\rangle \right| \leq (\rho_N \epsilon_N + \beta_N \epsilon_{\text{ratio}}) \|v'_j - v_j\| + \epsilon_{\text{ratio}}$, we control

the first sum.

$$\sum_{j \in [m]} |\overrightarrow{a_j}| |c'_j - c_j| \left| \left\langle \frac{v'_j}{N'_j}, \overline{x_i} \right\rangle \right| \leq \sum_{j \in [m]} |\overrightarrow{a_j}| |c'_j - c_j| \left( (\rho_N \epsilon_N + \beta_N \epsilon_{\text{ratio}}) \|v'_j - v_j\| + \epsilon_{\text{ratio}} \right)$$

$$\leq \|\overrightarrow{a}\| \|C' - C\| \left( \|V' - V\| (\rho_N \epsilon_N + \beta_N \epsilon_{\text{ratio}}) + \epsilon_{\text{ratio}} \right)$$

$$\leq R \left( \beta_V (\rho_N \epsilon_N + \beta_N \epsilon_{\text{ratio}}) + \epsilon_{\text{ratio}} \right)$$

$$\leq 5 \epsilon_N^2 R.$$

Using lemma D.1.3 to bound $\left| \left\langle \frac{v'_j}{N_j}, \overline{x_i} \right\rangle - \langle \overline{v_j}, \overline{x_i} \rangle \right|$ and the fact that $c = \overrightarrow{1}$, we control the

second sum.

$$\sum_{j \in [m]} |\overrightarrow{a_j}| |c_j| \left| \left\langle \frac{v'_j}{N'_j}, \overline{x_i} \right\rangle - \langle \overline{v_j}, \overline{x_i} \rangle \right| \leq \sum_{j \in [m]} |\overrightarrow{a_j}| \left( (\rho_N \epsilon_N + \beta_N \epsilon_{\text{ratio}}) \|v'_j - v_j\| + \left\| \frac{v_j}{N_j} - \overline{v_j} \right\| \right)$$

$$\leq \|\overrightarrow{a}\| \left( (\rho_N \epsilon_N + \beta_N \epsilon_{\text{ratio}}) \|V' - V\| + \sqrt{\sum_{j \in [m]} \left\| \frac{v_j}{N_j} - \overline{v_j} \right\|^2} \right)$$

$$\leq (\rho_N \epsilon_N + \beta_N \epsilon_{\text{ratio}}) \beta_V + \sqrt{m} \Delta_V$$

$$\leq 3 \epsilon_N^2 + \sqrt{m} \Delta_V.$$

Putting it altogether, we have

$$\left| \left\langle \overrightarrow{a}, \partial_a p_i(W_s) - \partial_a p_i(\overline{W^a}) \right\rangle \right| \leq 5 \epsilon_N^2 R + \sqrt{m} \Delta_V + 3 \epsilon_N^2. \tag{29}$$

By eqs. (28) and (29),

$$\left\langle \overrightarrow{a}, \partial_a p_i(W') \right\rangle = \left\langle \overrightarrow{a}, \partial_a p_i(\overline{W^a}) \right\rangle + \left\langle \overrightarrow{a}, \partial_a p_i(W') - \partial_a p_i(\overline{W^a}) \right\rangle$$

$$\geq \gamma_a \sqrt{m} - (5 \epsilon_N^2 R + \sqrt{m} \Delta_V + 3 \epsilon_N^2). \tag{30}$$

Similarly, note that

$$\left|\left\langle \overrightarrow{C}, \partial_c p_i(W_s) - \partial_c p_i(\overline{W^C})\right\rangle\right|$$

$$\leq \sum_{j\in[m]} |\overrightarrow{c_j}|\left|a_j o'_j|_{\overline{x_i}}\left\langle \frac{v'_j}{N'_j}, \overline{x_i}\right\rangle - a_j o_j|_{\overline{x_i}}\langle \overline{v_j}, \overline{x_i}\rangle\right|$$

$$\leq \sum_{j\in[m]} |\overrightarrow{c_j}||a_j - a_j|\left|o'_j|_{\overline{x_i}}\left\langle \frac{v'_j}{N'_j}, \overline{x_i}\right\rangle\right| + \sum_{j\in[m]} |\overrightarrow{c_j}||a_j|\left|o'_j|_{\overline{x_i}}\left\langle \frac{v'_j}{N'_j}, \overline{x_i}\right\rangle - o_j|_{\overline{x_i}}\langle \overline{v_j}, \overline{x_i}\rangle\right|.$$

Using lemma D.1.2 to get $\left|\left\langle \frac{v'_j}{N'_j}, \overline{x_i}\right\rangle\right| \leq (\rho_N \epsilon_N + \beta_N \epsilon_{\text{ratio}})\left\|v'_j - v_j\right\| + \epsilon_{\text{ratio}}$, we control the first sum.

$$\sum_{j\in[m]} |\overrightarrow{c_j}||a_j - a_j|\left|\left\langle \frac{v'_j}{N'_j}, \overline{x_i}\right\rangle\right| \leq \sum_{j\in[m]} |\overrightarrow{c_j}||a_j - a_j|\left((\rho_N \epsilon_N + \beta_N \epsilon_{\text{ratio}})\left\|v'_j - v_j\right\| + \epsilon_{\text{ratio}}\right)$$

$$\leq \left\|\overrightarrow{C}\right\|\left\|a' - a\right\|\left((\rho_N \epsilon_N + \beta_N \epsilon_{\text{ratio}})\left\|V' - V\right\| + \epsilon_{\text{ratio}}\right)$$

$$\leq R\left((\rho_N \epsilon_N + \beta_N \epsilon_{\text{ratio}})\beta_V + \epsilon_{\text{ratio}}\right)$$

$$\leq 5\epsilon_N^2 R.$$

Using lemma D.1.5 to bound $\left|o'_j|_{\overline{x_i}}\left\langle \frac{v'_j}{N'_j}, \overline{x_i}\right\rangle - o_j|_{\overline{x_i}}\langle \overline{v_j}, \overline{x_i}\rangle\right|$, we control the second sum.

$$\sum_{j\in[m]} |\overrightarrow{c_j}||a_j|\left|o'_j|_{\overline{x_i}}\left\langle \frac{v'_j}{N'_j}, \overline{x_i}\right\rangle - o_j|_{\overline{x_i}}\langle \overline{v_j}, \overline{x_i}\rangle\right|$$

$$\leq \sum_{j\in[m]} |\overrightarrow{c_j}||a_j|\left((\rho_N \epsilon_N + \beta_N \epsilon_{\text{ratio}})\left\|v'_j - v_j\right\| + \left\|\frac{v_j}{N_j} - \overline{v_j}\right\| + \epsilon_{\text{ratio}}\right)$$

$$\leq \left\|\overrightarrow{C}\right\|\|a\|\left((\rho_N \epsilon_N + \beta_N \epsilon_{\text{ratio}})\left\|V' - V\right\| + \sqrt{\sum_{j\in[m]}\left\|\frac{v_j}{N_j} - \overline{v_j}\right\|^2} + \epsilon_{\text{ratio}}\right)$$

$$\leq 10\epsilon_N^2 + 2\sqrt{m}\Delta_V.$$

Thus, we have

$$\left|\left\langle \overrightarrow{C}, \partial_c p_i(W_s) - \partial_c p_i(\overline{W^C})\right\rangle\right| \leq 5\epsilon_N^2 R + 2\sqrt{m}\Delta_V + 10\epsilon_N^2. \tag{31}$$

Therefore, by eqs. (28) and (31) we obtain

$$\left\langle \overrightarrow{C}, \partial_c p_i(W')\right\rangle = \left\langle \overrightarrow{C}, \partial_c p_i(\overline{W^C})\right\rangle + \left\langle \overrightarrow{C}, \partial_C p_i(W') - \partial_C p_i(\overline{W^C})\right\rangle$$

$$\geq \gamma_c\sqrt{m} - (5\epsilon_N^2 R + 2\sqrt{m}\Delta_V + 10\epsilon_N^2). \tag{32}$$

Therefore, by eqs. (30) and (32) we get that

$$\left\langle \overrightarrow{U}, \partial_U p_i(W')\right\rangle \geq \gamma\sqrt{m} - (10\epsilon_N^2 R + 3\sqrt{m}\Delta_V + 13\epsilon_N^2). \tag{33}$$

Since we have $R \leq \frac{\gamma^{1/3} m^{1/6}}{27\epsilon_N}$, $\sqrt{m}\Delta_V \leq \frac{\gamma\sqrt{m}}{16}$, and $m \geq \frac{2^{29}\epsilon_N^3 \ln^{3/2}(5n/\delta)}{\gamma^2}$, eq. (33) simplifies to

$$\left\langle \overrightarrow{U}, \partial_U p_i(W') \right\rangle \geq \gamma\sqrt{m} - \frac{\gamma\sqrt{m}}{2} = \frac{\gamma\sqrt{m}}{2}. \tag{34}$$

2. Discarding another $6\delta$ failure probability and invoking lemma D.2 and homogeneity,

$$
\begin{aligned}
&\left| \langle U, \partial_U p_i(W') \rangle - p_i(W') + \alpha \langle V - V', \partial_V p_i(W') \rangle \right| \\
&= \left| \langle a, \partial_a p_i(W') \rangle + \langle C' - C, \partial_c p_i(W') \rangle + \alpha \langle V - V', \partial_V p_i(W') \rangle \right| \\
&\leq \left| \langle a, \partial_a p_i(W') \rangle \right| + \left| \langle C' - C, \partial_c p_i(W') \rangle \right| + \alpha \left| \langle V - V', \partial_V p_i(W') \rangle \right| \\
&\leq R\frac{\gamma\sqrt{m}}{32} \\
&= r\frac{\gamma\sqrt{m}}{4}.
\end{aligned} \tag{35}
$$

From eqs. (35) and (34), we have that

$$
\begin{aligned}
&\left\langle \overline{U}, \partial_U p_i(W') \right\rangle - p_i(W') + \alpha \langle V - V', \partial_V p_i(W') \rangle \\
&= r\left\langle \overrightarrow{U}, \partial_U p_i(W') \right\rangle + \langle U, \partial_U p_i(W') \rangle - p_i(W') + \alpha \langle V - V', \partial_V p_i(W') \rangle \\
&\geq r(\frac{3\gamma\sqrt{m}}{4}) - r(\frac{\gamma\sqrt{m}}{4}) \\
&= r(\frac{\gamma\sqrt{m}}{2}).
\end{aligned}
$$

$\square$

# E  STOCHASTIC GRADIENT DESCENT

In this section, we will provide a proof of theorem 2.1.

We will start by providing a bound on the squared norm of the loss gradients. Then we will prove a martingale concentration inequality used to convert a training error guarantee into a test error guarantee. Finally, we will conclude with a proof of theorem 2.1

**Lemma E.1.** *Suppose weights $W = (a, c, V)$ where $(a, V) \sim \frac{\mathcal{N}_m}{\sqrt{m}} \times \frac{\mathcal{N}_{m \times d}}{\sqrt{d}}$, $c = \overrightarrow{1} \in \mathbb{R}^m$ are given. Consider radii $R, R_V$, and width m satisfying*

$$R \leq \frac{\gamma^{1/3} m^{1/6}}{27\epsilon_N}, \qquad R_V = \beta_V = \frac{1}{2}\epsilon_N, \qquad m \in \left[ \frac{2^{29}\epsilon_N^3 \ln^{3/2}(5n/\delta)}{\gamma^2}, \delta \exp(\frac{d}{8}) \right].$$

*Denote $\Gamma := \{W' \mid \|U' - U\| < R, \|V' - V\| < R_V\}$. For any $W' \in \Gamma$,*

$$\left\| \nabla_U p_i(W') \right\|^2 \leq 2^{11}\epsilon_N^4 m \quad and \quad \left\| \nabla_V p_i(W') \right\|^2 \leq 2^{16}\epsilon_N^4 R^4.$$

*proof of lemma E.1.* By lemma C.2, with probability at least $1 - \delta$, $\|V^\mathsf{T}\|_{2,\infty} \leq 2$. Given that $\beta_V = \frac{1}{2\epsilon_N}$, let $(\rho_N, \beta_N, \beta_\Pi, \epsilon_{\text{ratio}})$ be constants corresponding to $V$ in lemma B.2. Without loss of generality, assume $\epsilon_N \geq 1$. Then, we can simplify the constants:

$$\rho_N = 2, \qquad\qquad \beta_N = 2\epsilon_N^2, \qquad\qquad \beta_\Pi = 9\epsilon_N, \qquad\qquad \epsilon_{\text{ratio}} = 2\epsilon_N. \tag{36}$$

For any $i \in [n]$, expanding the square gives us

$$\left\| \nabla_U p_i(W_s) \right\|^2 = \underbrace{\left\| \sum_{j \in [m]} \sigma(c_j \left\langle \frac{v_j}{N(v_j)}, \overline{x_i} \right\rangle) e_j \right\|^2}_{(1)} + \underbrace{\left\| \sum_{j \in [m]} a_j o_j |_{\overline{x_i}} \left\langle \frac{v_j}{N(v_j)}, \overline{x_i} \right\rangle e_j \right\|^2}_{(2)}.$$

Since $W' \in \Gamma$, we have $\max \left( \left\| c' - c \right\|, \left\| a' - a \right\| \right) \leq \left\| U' - U \right\| \leq R$. Additionally, we have $\left| \frac{v'_j}{N_j} \right|^2 \leq 5^2 (2 + \beta_V)^2 \rho_N^2 \epsilon_N^2$ by eq. (12) which gives

$$(1) \leq \sum_{j \in [m]} (c'_j)^2 \left( \frac{v'_j}{N_j} \right)^2 \leq \left\| c' \right\|^2 \max_j \left| \frac{v'_j}{N_j} \right|^2 \leq (\left\| c' - c \right\| + \left\| c \right\|)^2 5^2 (2 + \beta_V)^2 \rho_N^2 \epsilon_N^2 \leq (R + \sqrt{m})^2 5^2 (2 + \beta_V)^2 (\rho_N \epsilon_N)^2.$$

$$(2) \leq \sum_{j \in [m]} (a'_j)^2 \frac{\left\| v'_j \right\|^2}{N(v'_j)^2} \leq \left\| a' \right\|^2 \max_j \left| \frac{v'_j}{N_j} \right|^2 \leq \left( \left\| a' - a \right\| + \left\| a \right\| \right)^2 \max_j \left| \frac{v'_j}{N_j} \right|^2 \leq (R+2)^2 5^2 (2 + \beta_V)^2 \rho_N^2 \epsilon_N^2.$$

Putting it altogether gives us

$$\left\| \nabla_U p_i(W_s) \right\|^2 = (1) + (2) \leq 2^{11} \epsilon^4 m.$$

Lemma B.2 establishes $\left\| \left( I - \frac{\Sigma v_j v_j^T}{N(v_j)^2} \right) \overline{x_i} \right\|^2 \leq \beta_\Pi$ which implies

$$\begin{aligned}
\left\| \nabla_V p_i(W_s) \right\|^2 &= \left\| \sum_{j \in [m]} \frac{a_j c_j o_j |_{\overline{x_i}}}{N(v_j)} \left( I - \frac{\Sigma v_j v_j^T}{N(v_j)^2} \right) e_j \overline{x_i}^\top \right\|^2 \\
&\leq \sum_{j \in [m]} \frac{(a'_j)^2 (c'_j)^2}{N(v'_j)^2} \left\| \left( I - \frac{\Sigma v_j v_j^T}{N(v_j)^2} \right) \overline{x_i} \right\|^2 \\
&\leq \left\| a' \right\|^2 \max_j \frac{\left| c'_j \right|^2}{N(v'_j)^2} \left\| \left( I - \frac{\Sigma v_j v_j^T}{N(v_j)^2} \right) \overline{x_i} \right\|^2 \\
&\leq (R+2)^2 (R+1)^2 (\rho_N \epsilon_N)^2 \beta_\Pi^2 \\
&\leq 2^{16} \epsilon_N^4 R^4.
\end{aligned}$$

$\square$

The following martingale concentration inequality is essentially from (Ji & Telgarsky, 2020, Lemma 4.3) and will be key tool for obtaining a test error guarantee.

**Lemma E.2** (martingale concentration). *Suppose $\{W_s\}_{s<t}$ is a sequence of weights where $W_s$ only depends on $\{(x_r, y_r)\}_{r<s}$. Denote $p$ as the dimension of $W_s$. Let $g : \mathbb{R}^d \times \{\pm 1\} \times \mathbb{R}^p \to \mathbb{R}$ satisfy $0 \leq g \leq 1$.*

*Then $\sum_{s<t} \mathbb{E}[g(x, y; W_s)] - g(x_s, y_s; W_s)$ is a martingale difference and with probability at least $1 - \delta$,*

$$\sum_{s<t} \mathbb{E}[g(x, y; W_s)] \leq 4 \sum_{s<t} g(x_s, y_s; W_s) + 4 \ln(\frac{1}{\delta}).$$

*Proof.* Let $\mathcal{F}_{s+1}$ be the $\sigma$-algebra generated by $(x_1, y_1), \ldots, (x_s, y_s)$ (i.e. $\mathcal{F}_{s+1} := \sigma((x_1, y_1), \ldots, (x_s, y_s)))$. Denote

$$X_s = \mathbb{E}_{(x,y)}\left[g(x, y; W_s)\right], \qquad Y_s = g(x_s, y_s; W_s), \qquad Z_s = X_s - Y_s.$$

Since $X_s = \mathbb{E}[Y_s \,|\, F_s]$,

$$\mathbb{E}[Z_s \,|\, F_s] = X_s - \mathbb{E}[Y_s \,|\, F_s] = 0.$$

Thus, $Z_s$ is a martingale difference.

Since $0 \le g \le 1$, we have that $Y_s, Z_s \le 1$. Therefore,

$$\mathbb{E}[Z_s^2 \,|\, \mathcal{F}_s] = \mathbb{E}[Y_s^2 - 2X_sY_s + X_s^2 \,|\, \mathcal{F}_s] = \mathbb{E}[Y_s^2 \,|\, \mathcal{F}_s] - X_s^2 \le \mathbb{E}[Y_s^2 \,|\, \mathcal{F}_s] \le \mathbb{E}[Y_s \,|\, \mathcal{F}_s] = X_s.$$

We can now invoke a variant of Freedman's inequality (Agarwal et al., 2014, Lemma 9)

$$\sum_{s<\tau} \mathbb{E}_{x,y}[g(x, y; W_s)] \le \sum_{s<\tau} g(x_s, y_s; W_s) + (e-2)\sum_{s<\tau} \mathbb{E}_{x,y}[g(x, y; W_s)] + \ln(1/\delta).$$

Rearranging and noticing $\frac{1}{3-e} \le 4$, we get the desired result.

$\square$

**Corollary E.3.** *With probability at least $1 - \delta$,*

$$\sum_{s<t} \mathbb{E}[\ell'(-p(x, y, W_s))|] \le 4\sum_{s<t}\left|\ell'(-p(x_s, y_s, W_s))\right| + 4\ln(1/\delta).$$

*Proof.* Let g be the function $x, y, W \to \left|\ell'(-p(x, y; W))\right|$. Since $|\ell'| \in [0, 1]$, we have that $0 \le g \le 1$. Now we can invoke lemma E.2 to get that with probability at least $1 - \delta$,

$$\sum_{s<t} \mathbb{E}[\ell'(-p(x, y, W_s))|] \le 4\sum_{s<t}\left|\ell'(-p(x_s, y_s, W_s))\right| + 4\ln(1/\delta).$$

$\square$

**Corollary E.4.** *With probability at least $1 - \delta$,*

$$\sum_{s<t} \mathbb{E}_{(x,y)}[\min\{\ell(-p(x, y; W_s)), \ln(2)\}] \le 4\sum_{s<t}\min\{\ell(-p(x_s, y_s; W_s)), \ln(2)\} + 4\ln(1/\delta).$$

*Proof.* Let g be the function $x, y, W \to \min\{\ell(-p(x, y; W)), \ln(2)\}$. Since $\ell \ge 0$ and $\ln(2) < 1$, we have that $0 \le g \le 1$. Now invoke lemma E.2 to get the desired result.

$\square$

**Theorem E.5** (theorem 2.1). *Suppose weights $W = (a, c, V)$ where $(a, V) \sim \frac{\mathcal{N}_m}{\sqrt{m}} \times \frac{\mathcal{N}_{m\times d}}{\sqrt{d}}$, $c = \overrightarrow{1} \in \mathbb{R}^m$ are given and assumption 1.2 holds, with corresponding $(\overrightarrow{a}, \overrightarrow{C}, \overrightarrow{V}) \in \mathbb{R}^m \times \mathbb{R}^m \times \mathbb{R}^{m\times d}$ and $\gamma = \gamma_a + \gamma_c$ given.*

*Suppose width satisfies*

$$m \in \left[\frac{2^{29}\epsilon_N^3 \ln^{3/2}(\frac{5nt}{\delta})}{\gamma^2}, \delta\exp(\frac{d}{8})\right],$$

*and we set the learning rate to be $\eta = \frac{1}{28\epsilon_N\gamma^{2/3}m^{1/3}}$.*

*With probability at least $1 - 20\delta$, the SGD iterates $(W_s)_{s\le t}$ satisfy*

- *an upper bound on parameter movement,*

$$\max_{s<t}\|U_s - U_0\| = \max_{s<t}\|(a_s, c_s) - (a_0, c_0)\| \le \frac{\gamma^{1/3}m^{1/6}}{27\epsilon_N},$$

$$\max_{s<t}\|V_s - V_0\| \le \frac{1}{2\epsilon_N},$$

- *a test error bound,*

$$\min_{s<t} \Pr(\text{sgn}(f(x; W_s)) \neq y) \leq \frac{2^{23}\epsilon_N^4}{\gamma^2 t} + \frac{4\ln(1/\delta)}{t}. \tag{37}$$

*Proof.* By lemma C.2, with probability at least $1 - 3\delta, \|V^\top\|_{2,\infty} \leq 2, \|a\| \leq 2$, and $\max_{j\in m} \frac{1}{\|v_j\|} \leq 5$. Given that $\beta_V = \frac{1}{2\epsilon_N}$, let $(\rho_N, \beta_N, \beta_\Pi, \epsilon_{\text{ratio}})$ be constants corresponding to $V$ in lemma B.2. Without loss of generality, assume $\epsilon_N \geq 1$. Then, we can simplify the constants:

$$\rho_N = 2, \qquad \beta_N = 2\epsilon_N^2, \qquad \beta_\Pi = 9\epsilon_N, \qquad \epsilon_{\text{ratio}} = 2\epsilon_N. \tag{38}$$

Set radii $R, R_V$, rescaling constant $\alpha$, and manifold $\Gamma$ as

$$R = \frac{\gamma^{1/3}m^{1/6}}{27\epsilon_N}, \quad R_V = \beta_V = \frac{1}{2}\epsilon_N, \quad \alpha = \frac{R^2}{\beta_V^2}, \quad \Gamma := \{W' \mid \|U' - U_0\| < R, \|V' - V_0\| < R_V\}.$$

Further, construct the following reference parameter

$$\overline{U} = (\overline{a}, \overline{C}) = r\overrightarrow{U} + U. \tag{39}$$

Consider the first time $\tau$ such that

$$\sqrt{\|U_\tau - U_0\|^2 + \alpha\|V_\tau - V_0\|^2} \geq R. \tag{40}$$

Assume contradictory that $\tau \leq t$ and note that for all time $s < \tau$, we have

$$\|U_s - U_0\| < R \quad \text{and} \quad \|V_s - V_0\| < R_V = \beta_V. \tag{41}$$

Since $W_s \in \Gamma$, discarding $7\delta$ failure probability, by lemma D.3,

$$\left\langle \overline{U}, \partial_U p_i(W') \right\rangle - p_i(W') + \alpha \left\langle V - V', \partial_V p_i(W') \right\rangle \geq r\left(\frac{\gamma\sqrt{m}}{2}\right) \geq \ln(5t). \tag{42}$$

Expanding the square,

$$\left\|U_{s+1} - \overline{U}\right\|^2 + \alpha\|V_{s+1} - V_0\|^2$$
$$= \left\|U_s - \overline{U}\right\|^2 + \alpha\|V_s - V_0\|^2 + 2\eta\left[\left\langle \partial\ell_s(W_s), \overline{U} - U_s\right\rangle + \alpha\left\langle \partial\ell_s(W_s), V_s - V_0\right\rangle\right] + \eta^2(1+\alpha)\left\|\partial\ell_s(W_s)\right\|^2. \tag{43}$$

Rearranging the terms and taking the sum over $s < \tau$, we have that

$$\Phi_D(\tau) := \left\|U_\tau - \overline{U}\right\|^2 - \left\|U_0 - \overline{U}\right\|^2 + \alpha\|V_\tau - V_0\|^2$$
$$\leq \underbrace{\sum_{s<\tau} 2\eta\ell_s'(W_s)\left[\left\langle \partial_U p_s(W_s), \overline{U} - U_s\right\rangle + \alpha\left\langle \partial_V p_s(W_s), V_s - V_0\right\rangle\right]}_{(A)}$$
$$+ \underbrace{\eta^2\sum_{s<\tau}\ell_s'(W_s)^2\left(\left\|\partial_U p_s(W_s)\right\|^2 + \alpha\left\|\partial_V p_s(W_s)\right\|^2\right)}_{(B)}. \tag{44}$$

By lemma D.3, convexity of $\ell$, and eq. (59), we can control the inner product term (A);

$$
\begin{aligned}
(A) &= 2\eta \sum_{s<\tau} \frac{\ell'_s(W_s)}{n} \left[ \left\langle \overline{U}, \partial_U p_i(W') \right\rangle + \alpha \left\langle V_0 - V', \partial_V p_i(W') - p_i(W') \right\rangle \right] \\
&\leq \frac{2\eta}{n} \sum_{s<\tau} \ell \left( \left\langle \overline{U}, \partial_U p_i(W') \right\rangle + \alpha \left\langle V_0 - V', \partial_V p_i(W') \right\rangle \right) - \ell(W_s) \\
&\leq \frac{2\eta}{n} \sum_{s<\tau} \ell(\ln(5t)) - \ell_i(W_s) \\
&\leq \frac{2\eta}{n} \sum_{s<\tau} \frac{1}{5t} - \ell_i(W_s) \\
&\leq 2\eta \left( \frac{\tau}{5t} - \sum_{s<\tau} \ell_s(W_s) \right) \\
&\leq \frac{2\eta}{5}.
\end{aligned}
\tag{45}
$$

Then by eq. (61), the rescaled Euclidean potential $\Phi_D$ simplifies to

$$
\begin{aligned}
& \left\| U_\tau - \overline{U} \right\|^2 - \left\| U_0 - \overline{U} \right\|^2 + \alpha \| V_\tau - V_0 \|^2 \\
& \leq \frac{2\eta}{5} + \underbrace{\eta^2 \sum_{s<\tau} \left( \left\| \ell'_s(W_s) \partial_U p_s(W_s) \right\|^2 + \alpha \left\| \ell'_s(W_s) \partial_V p_s(W_s) \right\|^2 \right)}_{(B)}.
\end{aligned}
\tag{46}
$$

By lemma E.1 , we can control the sum containing the squared gradient norm terms:

$$
(B) = \eta^2 \sum_{s<\tau} \left( \left\| \ell'_s(W_s) \partial_U p_s(W_s) \right\|^2 + \alpha \left\| \ell'_s(W_s) \partial_V p_s(W_s) \right\|^2 \right)
\tag{47}
$$

$$
\leq \eta^2 (2^{11} \epsilon_N^4 m + 2^{16} \alpha \epsilon_N^4 R^4) \sum_{s<\tau} \left( \left| \ell'_s(W_s) \right| \right)
\tag{48}
$$

$$
\leq \eta^2 (2^{11} \epsilon_N^4 m + 2^{16} \epsilon_N^6 R^6) \sum_{s<\tau} \left( \left| \ell'_s(W_s) \right| \right).
\tag{49}
$$

To control $\sum_{s<\tau} \left( \left| \ell'_s(W_s) \right| \right)$, we use the perceptron argument

$$
\begin{aligned}
\| U_\tau - U_0 \| &= \sup_{\|U\| \leq 1} \langle U, U_\tau - U_0 \rangle \\
&\geq \left\langle \overline{U}, U_\tau - U_0 \right\rangle \\
&\geq \left\langle \overline{U}, \eta \sum_{s<\tau} -\ell'_s(W_s) \left\langle \overline{U}, \partial_U p_i(W_s) \right\rangle \right\rangle \\
&\geq \eta \frac{\gamma \sqrt{m}}{4} \sum_{s<\tau} \left| \ell'_s(W_s) \right|.
\end{aligned}
$$

Rearranging gives

$$
\sum_{s<\tau} \left| \ell'_s(W_s) \right| \leq \frac{4 \| U_\tau - U_0 \|}{\eta \gamma \sqrt{m}}.
\tag{50}
$$

Combining eqs. (47) and (50) gives us

$$
\Phi_D(\tau) \leq \frac{2\eta}{5} + \eta \left( \frac{2^{13} \epsilon_N^4 \sqrt{m}}{\gamma} + \frac{2^{18} \epsilon_N^6 R^6}{\gamma \sqrt{m}} \right) \| U_\tau - U_0 \|.
\tag{51}
$$

Now observe

$$
\begin{aligned}
\Phi_D(\tau) &= \left\|U_\tau - \overline{U}\right\|^2 - \left\|U_0 - \overline{U}\right\| + \alpha\|V_\tau - V_0\|^2 \\
&\geq \|U_\tau - U_0\|^2 + \alpha\|V_\tau - V_0\|^2 - 2\|U_\tau - U_0\|\left\|\overline{U} - U_0\right\| \\
&\geq \|U_\tau - U_0\|^2 + \alpha\|V_\tau - V_0\|^2 - 2r\left\|(a_\tau, C_\tau) - (a, C)\right\|.
\end{aligned}
$$

From the preceding quadratic inequality and eq. (62), we obtain

$$
\begin{aligned}
\|U_\tau - U_0\|^2 + \alpha\|V_\tau - V_0\|^2 &\leq 2r\left\|(a_\tau, C_\tau) - (a, C)\right\| + \Phi_D(\tau) \\
&\leq 2r\left\|(a_\tau, C_\tau) - (a, C)\right\| + \frac{2\eta}{5} + \eta\left(\frac{2^{13}\epsilon_N^4\sqrt{m}}{\gamma} + \frac{2^{18}\epsilon_N^6 R^6}{\gamma\sqrt{m}}\right)\|U_\tau - U_0\|.
\end{aligned}
$$

Dividing both sides by $\sqrt{\|U_\tau - U_0\|^2 + \alpha\|V_\tau - V_0\|^2}$ gives us

$$
\begin{aligned}
\sqrt{\|U_\tau - U_0\|^2 + \alpha\|V_\tau - V_0\|^2} &\leq \frac{2r\|U_\tau - U_0\|}{\sqrt{\|U_\tau - U_0\|^2 + \alpha\|V_\tau - V_0\|^2}} + \frac{\eta\left(\frac{2}{5} + \frac{2^{13}\epsilon_N^4\sqrt{m}}{\gamma} + \frac{2^{18}\epsilon_N^6 R^6}{\gamma\sqrt{m}}\right)\|U_\tau - U_0\|}{\sqrt{\|U_\tau - U_0\|^2 + \alpha\|V_\tau - V_0\|^2}} \\
&\leq 2r + \eta\left(\frac{2}{5} + \frac{2^{13}\epsilon_N^4\sqrt{m}}{\gamma} + \frac{2^{18}\epsilon_N^6 R^6}{\gamma\sqrt{m}}\right) \\
&\leq R/2 + R/20 + R/8 + R/8 \\
&< R.
\end{aligned}
$$

This a contradiction and thus for all time $s \leq t$, we have that

$$
\sqrt{\|U_s - U_0\|^2 + \alpha\|V_s - V_0\|^2} < R
$$

By eq. (50) and since $R = \frac{\gamma^{1/3}m^{1/6}}{27\epsilon_N}$, we get

$$
\sum_{s<\tau}|\ell_s'(W_s)| \leq \frac{4\|U_\tau - U_0\|}{\eta\gamma\sqrt{m}} \leq \frac{4R}{\eta\gamma\sqrt{m}} \leq \frac{2^{22}\epsilon_N^4}{\gamma^2}.
$$

Using the fact that $\mathbf{1}_{y\neq\mathrm{sgn}(f(x;W_s))} \leq 2|\ell'(p(x,y,W_s))|$ and corollary E.3, we get

$$
\begin{aligned}
t\min_{s<t}\Pr(y \neq \mathrm{sgn}(f(x;W_s))) &\leq \sum_{s<t}\Pr(y \neq \mathrm{sgn}(f(x;W_s))) \\
&= \sum_{s<t}\mathbb{E}[\mathbf{1}_{\{y\neq\mathrm{sgn}(f(x;W_s))\}}] \\
&\leq \frac{1}{2}\sum_{s<t}\mathbb{E}_{(x,y)}[|\ell'(-p(x,y,W_s))|] \\
&\leq 2\sum_{s<t}|\ell'(-p(x_s,y_s,W_s))| + 4\ln(1/\delta) \\
&\leq \frac{2^{23}\epsilon_N^4}{\gamma^2} + 4\ln(1/\delta).
\end{aligned}
$$

Dividing by $t$ gives the desired results.

$\square$

## F   GRADIENT DESCENT

This section will be devoted to proving theorem 2.2. We first start by establishing some properties regarding Rademacher complexity.

### F.1 RADEMACHER COMPLEXITY

In this section, we provide an upper bound for the Rademacher complexity of batch-normalized networks which will be used to obtain test error guarantees for networks trained by GF.

We will utilize many of the techniques used in the proof of Telgarsky (2022, lemma C.5) though some care is needed to take advantage of the additional constraint embedded in the hypothesis class: $\|V - V_0\| \le \rho$ for some $\rho > 0$.

Before presenting the lemma, we develop some notation. Let $\mathcal{G} \subset \mathbb{R}^d \to \mathbb{R}$ and $\mathcal{X} = \{(x_i, y_i)\}_{i \in [n]}$. Define $*$ be the operation such that

$$\mathcal{G} * \mathcal{X} := \{(\phi(x_1), \dots, \phi(x_n)) \mid \phi \in \mathcal{G}\}$$

In standard expositions for Rademacher complexity (e.g. Shalev-Shwartz & Ben-David (2014)) the notiation $\circ$ is used in place of $*$ but we shall favor the usage of $*$ as $\circ$ will denote another operation as indicated in lemma F.2.

**Lemma F.1.** *Suppose data $\mathcal{X} = \{(x_i, y_i)\}_{i \in [n]}$ is given. Consider the hypothesis class*

$$\mathcal{H} = \left\{ (x, y) \to y \sum_{j \in [m]} a_j \sigma(c_j \left\langle \frac{v_j}{N(v_j)}, x - \mu \right\rangle) : \sum_{j \in [m]} \left| \frac{a_j c_j}{N(v_j)} \right| \le B, \|V - V_0\| \le \rho \right\}.$$

*Then*

$$\mathrm{Rad}\left(\mathcal{H} * \mathcal{X}\right) \le \frac{4B\rho \left\| \overline{X} \right\|}{n} \le \frac{4B\rho}{\sqrt{n}}.$$

*Proof.* Define

$$\widetilde{a}_j = \mathrm{sgn}(a_j) \quad \text{and} \quad \widetilde{c}_j = \mathrm{sgn}(c_j).$$

Let sconv denote the symmetric convex hull. Note

$$\mathcal{H} = \left\{ (x, y) \to y \sum_{j \in [m]} a_j \sigma(c_j \left\langle \frac{v_j}{N(v_j)}, x - \mu \right\rangle) : \sum_{j \in [m]} \left| \frac{a_j c_j}{N(v_j)} \right| \le B, \|V - V_0\| \le \rho \right\}$$

$$= \left\{ (x, y) \to y \sum_{k \in [n]} \left| \frac{a_k c_k}{N(v_k)} \right| \sum_{j \in [m]} \frac{|a_j c_j| \widetilde{a}_j}{\sum_{k \in [n]} \left| \frac{a_k c_k}{N(v_k)} \right|} \sigma(\widetilde{c}_j \left\langle v_j, x - \mu \right\rangle) : \sum_{j \in [m]} \left| \frac{a_j c_j}{N(v_j)} \right| \le B, \|V - V_0\| \le \rho \right\}$$

$$\subseteq B \left\{ (x, y) \to y \sum_{j \in [m]} p_j \sigma(\widetilde{c}_j \left\langle u_j, x - \mu \right\rangle) : p \in \mathbb{R}^m, \|p\|_1 \le 1, \widetilde{c}_j \in \{\pm 1\}, \|V - V_0\| \le \rho \right\}$$

$$\subseteq B \, \mathrm{sconv}\left( \left\{ (x, y) \to y \sigma(\widetilde{c}_j \left\langle v_j, x - \mu \right\rangle) : \|V - V_0\| \le \rho, \widetilde{c}_j \in \{\pm 1\} \right\} \right).$$

Thus by standard facts from Rademacher Calculus (Shalev-Shwartz & Ben-David, 2014),

$$n \cdot \mathrm{Rad}\left( B \, \mathrm{sconv}\left( \left\{ (x, y) \to y \sigma(\widetilde{c}_j \left\langle v_j, x - \mu \right\rangle) : \|V - V_0\| \le \rho, \widetilde{c}_j \in \{\pm 1\} \right\} * \mathcal{X} \right) \right)$$

$$\le 2nB \mathrm{Rad}\left( \left\{ (x, y) \to y \sigma(\widetilde{c}_j \left\langle v_j, x - \mu \right\rangle) : \|V - V_0\| \le \rho, \widetilde{c}_j \in \{\pm 1\} \right\} * \mathcal{X} \right)$$

$$\le 2nB \mathrm{Rad}\left( \left\{ (x, y) \to y \widetilde{c}_j \left\langle v_j, x - \mu \right\rangle : \|V - V_0\| \le \rho, \widetilde{c}_j \in \{\pm 1\} \right\} * \mathcal{X} \right). \tag{52}$$

Let $\mathcal{G} := \left\{ (x, y) \to y \left\langle v_j, x - \mu \right\rangle : \|V - V_0\| \le \rho, j \in [m] \right\}$.

Then eq. (52) can be rewritten as

$$2nB\text{Rad}\left(\{G \cup -G\}*\mathcal{X}\right) \leq 4nB\text{Rad}\left(G*\mathcal{X}\right)$$

$$= 4B\mathbb{E}_\epsilon \sup_{j\in[m]} \sup_{\substack{\|v-v_{0j}\|\leq\rho_j, \\ \sum_{k\in[m]}\rho_k^2=\rho^2}} \sum_{i\in[n]} \epsilon_i \langle v, y_i\overline{x_i}\rangle$$

$$= 4B\mathbb{E}_\epsilon \sup_{j\in[m]} \sup_{\substack{\|v-v_{0j}\|\leq\rho_j, \\ \sum_{k\in[m]}\rho_k^2=\rho^2}} \sum_{i\in[n]} \epsilon_i \langle v-v_{0j}, y_i\overline{x_i}\rangle$$

$$= 4B\mathbb{E}_\epsilon \sup_{j\in[m]} \sup_{\substack{\|u_j\|\leq\rho_j, \\ \sum_{k\in[m]}\rho_k^2=\rho^2}} \sum_{i\in[n]} \epsilon_i \langle u_j, y_i\overline{x_i}\rangle$$

$$= 4B\mathbb{E}_\epsilon \sup_{j\in[m]} \sup_{\rho_j:\sum_{k\in[m]}\rho_k^2=\rho^2} \rho_j \left\|\sum_{i\in[n]} \epsilon_i y_i\overline{x_i}\right\|$$

$$= 4B\mathbb{E}_\epsilon \rho \left\|\sum_{i\in[n]} \epsilon_i y_i\overline{x_i}\right\|$$

$$\leq 4B\rho \left\|\overline{X}\right\|_F.$$

where the last inequality follows from the fact that

$$\mathbb{E}_\epsilon \left\|\sum_{i\in[n]} \epsilon_i y_i\overline{x_i}\right\| \leq \sqrt{\mathbb{E}_\epsilon \left\|\sum_{i\in[n]} \epsilon_i y_i\overline{x_i}\right\|^2} = \sqrt{\mathbb{E}_\epsilon \sum_{i\in[n]} \|\overline{x_i}\|^2 + \mathbb{E}_\epsilon \sum_{i\neq j} \epsilon_i\epsilon_j \langle y_i\overline{x_i}, y_j\overline{x_j}\rangle} = \left\|\overline{X}\right\|_F.$$

$\square$

We will need a variation of the contraction lemma (also known as the peeling lemma) (Shalev-Shwartz & Ben-David, 2014, lemma 26.9).

**Lemma F.2.** *For each $i \in [n]$, let $\phi_i : \mathbb{R} \to \mathbb{R}$ be $\rho$ - lipschitz on $K_i \subset \mathbb{R}$. For $a \in \mathbb{R}^n$, let $\phi(a) = \left(\phi_1(a_1), \dots, \phi_n(a_n)\right)$. Denote $\phi \circ A = \{\phi(a) : a \in A\}$. Suppose additionally we have $A \subset K_1 \times \cdots \times K_n$. Then*

$$\text{Rad}(\phi \circ A) \leq \rho\,\text{Rad}(A)$$

The proof of lemma F.2 follows directly from the proof of Shalev-Shwartz & Ben-David (2014, lemma 26.9) and the observation that one only needs $\phi_i$ to be $\rho$-lipschitz on $\pi_i(A)$ where $\pi_i(a) = a_i$ for any vector $a \in \mathbb{R}^n$. Thus, we defer the full proof to Shalev-Shwartz & Ben-David (2014, lemma 26.9).

**Lemma F.3.** *In this section, we will control the squared gradient norm.*

$$\left\|\nabla_U \widehat{\mathcal{R}}(W_s)\right\|^2 \leq 175\epsilon_N^4 m \sum_{i\in[n]} \frac{|\ell_i'(W_s)|}{n}$$

$$\left\|\nabla_V \widehat{\mathcal{R}}(W_s)\right\|^2 \leq 8100\epsilon_N^4 R^4 \sum_{i\in[n]} \frac{|\ell_i'(W_s)|}{n}$$

For any $i \in [n]$, we have that

$$\left\|\nabla_U p_i(W_s)\right\|^2 = \underbrace{\left\|\sum_{j\in[m]} \sigma(c_j\left\langle \frac{v_j}{N(v_j)}, \overline{x_i}\right\rangle)e_j\right\|^2}_{(1)} + \underbrace{\left\|\sum_{j\in[m]} a_j o_j|_{\overline{x_i}}\left\langle \frac{v_j}{N(v_j)}, \overline{x_i}\right\rangle e_j\right\|^2}_{(2)}$$

Using lemma B.2

$$(1) \leq \sum_{j \in [m]} (c_j')^2 \left( \frac{v_j'}{N_j} \right)^2 \leq \|c'\|^2 \max_j \left| \frac{v_j'}{N_j} \right|^2 \leq (\|c' - c\| + \|c\|)^2 (2 + \beta_V)^2 \rho_N^2 \epsilon_N^2 \leq (R + \sqrt{m})^2 (2 + \beta_V)^2 (\rho_N \epsilon_N)^2$$

$$(2) \leq \sum_{j \in [m]} (a_j')^2 \frac{\|v_j'\|^2}{N(v_j')^2} \leq \|a'\|^2 \max_j \left| \frac{v_j'}{N_j} \right|^2 \leq \left( \|a' - a\| + \|a\| \right)^2 \max_j \left| \frac{v_j'}{N_j} \right|^2 \leq (R + 2)^2 (2 + \beta_V)^2 \rho_N^2 \epsilon_N^2$$

Therefore,

$$\left\| \nabla_U p_i(W_s) \right\|^2 = (1) + (2) \leq 175 \epsilon^4 m$$

By lemma B.2.1 and lemma B.2.3

$$\left\| \nabla_V p_i(W_s) \right\|^2 = \left\| \sum_{j \in [m]} \frac{a_j c_j o_j|_{\overline{x_i}}}{N(v_j)} \left( I - \frac{\Sigma v_j v_j^T}{N(v_j)^2} \right) e_j \overline{x_i}^\top \right\|^2$$

$$\leq \sum_{j \in [m]} \frac{(a_j')^2 (c_j')^2}{N(v_j')^2} \left\| \left( I - \frac{\Sigma v_j v_j^T}{N(v_j)^2} \right) \overline{x_i} \right\|^2$$

$$\leq \|a'\|^2 \max_j \frac{|c_j'|^2}{N(v_j')^2} \left\| \left( I - \frac{\Sigma v_j v_j^T}{N(v_j)^2} \right) \overline{x_i} \right\|^2$$

$$\leq (R + 2)^2 (R + 1)^2 (\rho_N \epsilon_N)^2 \beta_\Pi^2$$

$$\leq 8100 \epsilon_N^4 R^4$$

In addition, note that since $|\ell'| \leq 1$, we have that $\sum_{i \in [n]} \frac{|\ell_i'(W_s)|}{n} \leq 1$ which implies

$$\left( \sum_{i \in [n]} \frac{|\ell_i'(W_s)|}{n} \right)^2 \leq \sum_{i \in [n]} \frac{|\ell_i'(W_s)|}{n} \tag{53}$$

Consequently, by eq. (53)

$$\left\| \nabla_U \widehat{\mathcal{R}}(W_s) \right\|^2 = \left\| \sum_{i \in [n]} \ell_i'(W_s) \nabla_U p_i(W_s) \right\|^2$$

$$\leq \left( \sum_{i \in [n]} \frac{|\ell_i'(W_s)|}{n} \right)^2 \max_{k \in [n]} \left\| \nabla_U p_k(W_s) \right\|^2$$

$$\leq \left( \sum_{i \in [n]} \frac{|\ell_i'(W_s)|}{n} \right) 175 \epsilon_N^4 m$$

Repeating the same argument, we get that

$$\left\| \nabla_V \widehat{\mathcal{R}}(W_s) \right\|^2 = \left\| \sum_{i \in [n]} \ell_i'(W_s) \nabla_V p_i(W_s) \right\|^2 \leq 8100 \epsilon_N^4 R^4 \sum_{i \in [n]} \frac{|\ell_i'(W_s)|}{n}$$

**Theorem F.4** (theorem 2.2). *Suppose weights $W = (a, c, V)$ where $(a, V) \sim \frac{\mathcal{N}_m}{\sqrt{m}} \times \frac{\mathcal{N}_{m \times d}}{\sqrt{d}}$, $c = \overrightarrow{1} \in \mathbb{R}^m$ are given and assumption 1.2 holds, with corresponding $(\overrightarrow{a}, \overrightarrow{C}, \overrightarrow{V}) \in \mathbb{R}^m \times \mathbb{R}^m \times \mathbb{R}^{m \times d}$ and $\gamma = \gamma_a + \gamma_c$ given.*

*Suppose width satisfies*

$$m \in \left[ \frac{2^{29} \epsilon_N^3 \ln^{3/2}(\frac{5nt}{\delta})}{\gamma^2}, \, \delta \exp(\frac{d}{8}) \right],$$

*and we set the learning rate to be $\eta = \frac{1}{28 \epsilon_N \gamma^{2/3} m^{1/3}}$.*

*With probability at least $1 - 20\delta$, the SGD iterates $(W_s)_{s \leq t}$ satisfy*

- *an upper bound on parameter movement,*

$$\max_{s < t} \|U_s - U_0\| = \max_{s < t} \|(a_s, c_s) - (a_0, c_0)\| \leq \frac{\gamma^{1/3} m^{1/6}}{27 \epsilon_N},$$

$$\max_{s < t} \|V_s - V_0\| \leq \frac{1}{2 \epsilon_N},$$

- *a test error bound,*

$$\min_{s < t} \Pr(\operatorname{sgn}(f(x; W_s)) \neq y) \leq \frac{2^{23} \epsilon_N^4}{\gamma^2 t} + \frac{4 \ln(1/\delta)}{t}. \tag{54}$$

*Proof.* By lemma C.2, with probability at least $1 - 3\delta$, $\|V^\top\|_{2,\infty} \leq 2$, $\|a\| \leq 2$, and $\max_{j \in m} \frac{1}{\|v_j\|} \leq 5$. Given that $\beta_V = \frac{1}{2\epsilon_N}$, let $(\rho_N, \beta_N, \beta_\Pi, \epsilon_{\text{ratio}})$ be constants corresponding to $V$ in lemma B.2. Without loss of generality, assume $\epsilon_N \geq 1$. Then, we can simplify the constants:

$$\rho_N = 2, \qquad \beta_N = 2\epsilon_N^2, \qquad \beta_\Pi = 9\epsilon_N, \qquad \epsilon_{\text{ratio}} = 2\epsilon_N. \tag{55}$$

Set radii $R, R_V$, rescaling constant $\alpha$, and manifold $\Gamma$ as

$$R = \frac{\gamma^{1/3} m^{1/6}}{27 \epsilon_N}, \quad R_V = \beta_V = \frac{1}{2} \epsilon_N, \quad \alpha = \frac{R^2}{\beta_V^2}, \quad \Gamma := \{W' \mid \|U' - U_0\| < R, \|V' - V_0\| < R_V\}.$$

Further, construct the following reference parameter

$$\overline{U} = (\overline{a}, \overline{C}) = r\overrightarrow{U} + U. \tag{56}$$

Consider the first time $\tau$ such that

$$\sqrt{\|U_\tau - U_0\|^2 + \alpha \|V_\tau - V_0\|^2} \geq R. \tag{57}$$

Assume contradictory that $\tau \leq t$ and note that for all time $s < \tau$, we have

$$\|U_s - U_0\| < R \quad \text{and} \quad \|V_s - V_0\| < R_V = \beta_V. \tag{58}$$

Since $W_s \in \Gamma$, discarding $7\delta$ failure probability, by lemma D.3,

$$\left\langle \overline{U}, \partial_U p_i(W') \right\rangle - p_i(W') + \alpha \left\langle V - V', \partial_V p_i(W') \right\rangle \geq r(\frac{\gamma \sqrt{m}}{2}) \geq \ln(5t). \tag{59}$$

Expanding the square,

$$\left\| U_{s+1} - \overline{U} \right\|^2 + \alpha \|V_{s+1} - V_0\|^2 = \left\| U_s - \overline{U} \right\|^2 + \alpha \|V_s - V_0\|^2$$
$$+ 2\eta \left[ \sum_{i \in [n]} \frac{\ell_i'(W_s)}{n} \left\langle \partial_U p_i(W_s), \overline{U} - U_s \right\rangle + \alpha \sum_{i \in [n]} \frac{\ell_i'(W_s)}{n} \left\langle \partial_V p_i(W_s), V_s - V_0 \right\rangle \right]$$
$$+ \eta^2 \left( \left\| \sum_{i \in [n]} \partial_U \ell_i(W_s) \right\|^2 + \alpha \left\| \sum_{i \in [n]} \partial_V \ell_i(W_s) \right\|^2 \right)$$

Rearranging the terms and taking the sum over $s < \tau$, we have that

$$
\begin{aligned}
\Phi_D(\tau) :=& \left\| U_\tau - \overline{U} \right\|^2 - \left\| U_0 - \overline{U} \right\|^2 + \alpha \| V_\tau - V_0 \|^2 \\
\leq & \underbrace{2\eta \sum_{s<\tau} \sum_{i\in[n]} \frac{\ell_i'(W_s)}{n} \left[ \left\langle \partial_U p_i(W_s), \overline{U} - U_s \right\rangle + \alpha \left\langle \partial_V p_i(W_s), V_s - V_0 \right\rangle \right]}_{(A)} \\
& + \underbrace{\eta^2 \sum_{s<\tau} \left( \left\| \sum_{i\in[n]} \frac{\partial_U \widehat{\mathcal{R}}(W_s)}{n} \right\|^2 + \alpha \left\| \sum_{i\in[n]} \partial_V \widehat{\mathcal{R}}(W_s) \right\|^2 \right)}_{(B)}.
\end{aligned}
\tag{60}
$$

By lemma D.3, convexity of $\ell$, and eq. (59), we can control the inner product term (A);

$$
\begin{aligned}
(A) =& 2\eta \sum_{s<\tau} \frac{\ell_i'(W_s)}{n} \left[ \left\langle \overline{U}, \partial_U p_i(W') \right\rangle + \alpha \left\langle V_0 - V', \partial_V p_i(W') - p_i(W') \right\rangle \right] \\
\leq & \frac{2\eta}{n} \sum_{s<\tau} \ell \left( \left\langle \overline{U}, \partial_U p_i(W') \right\rangle + \alpha \left\langle V_0 - V', \partial_V p_i(W') \right\rangle \right) - \ell(W_s) \\
\leq & \frac{2\eta}{n} \sum_{s<\tau} \ell(\ln(5t)) - \ell_i(W_s) \\
\leq & \frac{2\eta}{n} \sum_{s<\tau} \frac{1}{5t} - \ell_i(W_s) \\
\leq & 2\eta \left( \frac{\tau}{5t} - \sum_{s<\tau} \ell_s(W_s) \right) \\
\leq & \frac{2\eta}{5}.
\end{aligned}
\tag{61}
$$

Then by eq. (61), the rescaled Euclidean potential $\Phi_D$ simplifies to

$$
\begin{aligned}
& \left\| U_\tau - \overline{U} \right\|^2 - \left\| U_0 - \overline{U} \right\|^2 + \alpha \| V_\tau - V_0 \|^2 \\
& \leq \frac{2\eta}{5} + \underbrace{\eta^2 \sum_{s<\tau} \left( \left\| \sum_{i\in[n]} \frac{\partial_U \widehat{\mathcal{R}}(W_s)}{n} \right\|^2 + \alpha \left\| \sum_{i\in[n]} \partial_V \widehat{\mathcal{R}}(W_s) \right\|^2 \right)}_{(B)}.
\end{aligned}
\tag{62}
$$

By lemma E.1 , we can control the sum containing the squared gradient norm terms:

$$
(B) = \eta^2 \sum_{s<\tau} \left( 175 \epsilon_N^4 m \sum_{i\in[n]} \frac{|\ell_i'(W_s)|}{n} + \alpha 8100 \epsilon_N^4 R^4 \sum_{i\in[n]} \frac{|\ell_i'(W_s)|}{n} \right)
\tag{63}
$$

$$
\leq \eta^2 (2^{11} \epsilon_N^4 m + 2^{16} \alpha \epsilon_N^4 R^4) \sum_{s<\tau} \sum_{i\in[n]} \frac{|\ell_i'(W_s)|}{n}
\tag{64}
$$

$$
\leq \eta^2 (2^{11} \epsilon_N^4 m + 2^{16} \epsilon_N^6 R^6) \sum_{s<\tau} \sum_{i\in[n]} \frac{|\ell_i'(W_s)|}{n}.
\tag{65}
$$

To control $\sum_{s<\tau} \sum_{i\in[n]} \frac{|\ell_i'(W_s)|}{n}$, we use the perceptron argument

$$
\begin{aligned}
\|U_\tau - U_0\| &= \sup_{\|U\| \leq 1} \langle U, U_\tau - U_0 \rangle \\
&\geq \left\langle \overline{U}, U_\tau - U_0 \right\rangle \\
&\geq \left\langle \overline{U}, \eta \sum_{s<\tau} -\ell_i'(W_s) \left\langle \overline{U}, \partial_U p_i(W_s) \right\rangle \right\rangle \\
&\geq \eta \frac{\gamma \sqrt{m}}{4} \sum_{s<\tau} \left| \ell_i'(W_s) \right|.
\end{aligned}
$$

Rearranging gives

$$
\sum_{s<\tau} \left| \ell_i'(W_s) \right| \leq \frac{4\|U_\tau - U_0\|}{\eta \gamma \sqrt{m}}. \tag{66}
$$

Combining eqs. (63) and (66) gives us

$$
\Phi_D(\tau) \leq \frac{2\eta}{5} + \eta \left( \frac{2^{13} \epsilon_N^4 \sqrt{m}}{\gamma} + \frac{2^{18} \epsilon_N^6 R^6}{\gamma \sqrt{m}} \right) \|U_\tau - U_0\|. \tag{67}
$$

Now observe

$$
\begin{aligned}
\Phi_D(\tau) &= \left\| U_\tau - \overline{U} \right\|^2 - \left\| U_0 - \overline{U} \right\| + \alpha \|V_\tau - V_0\|^2 \\
&\geq \|U_\tau - U_0\|^2 + \alpha \|V_\tau - V_0\|^2 - 2\|U_\tau - U_0\| \left\| \overline{U} - U_0 \right\| \\
&\geq \|U_\tau - U_0\|^2 + \alpha \|V_\tau - V_0\|^2 - 2r \left\| (a_\tau, C_\tau) - (a, C) \right\|.
\end{aligned}
$$

From the preceding quadratic inequality and eq. (62), we obtain

$$
\begin{aligned}
\|U_\tau - U_0\|^2 + \alpha \|V_\tau - V_0\|^2 &\leq 2r \left\| (a_\tau, C_\tau) - (a, C) \right\| + \Phi_D(\tau) \\
&\leq 2r \left\| (a_\tau, C_\tau) - (a, C) \right\| + \frac{2\eta}{5} + \eta \left( \frac{2^{13} \epsilon_N^4 \sqrt{m}}{\gamma} + \frac{2^{18} \epsilon_N^6 R^6}{\gamma \sqrt{m}} \right) \|U_\tau - U_0\|.
\end{aligned}
$$

Dividing both sides by $\sqrt{\|U_\tau - U_0\|^2 + \alpha \|V_\tau - V_0\|^2}$ gives us

$$
\begin{aligned}
\sqrt{\|U_\tau - U_0\|^2 + \alpha \|V_\tau - V_0\|^2} &\leq \frac{2r\|U_\tau - U_0\|}{\sqrt{\|U_\tau - U_0\|^2 + \alpha \|V_\tau - V_0\|^2}} + \frac{\eta \left( \frac{2}{5} + \frac{2^{13} \epsilon_N^4 \sqrt{m}}{\gamma} + \frac{2^{18} \epsilon_N^6 R^6}{\gamma \sqrt{m}} \right) \|U_\tau - U_0\|}{\sqrt{\|U_\tau - U_0\|^2 + \alpha \|V_\tau - V_0\|^2}} \\
&\leq 2r + \eta \left( \frac{2}{5} + \frac{2^{13} \epsilon_N^4 \sqrt{m}}{\gamma} + \frac{2^{18} \epsilon_N^6 R^6}{\gamma \sqrt{m}} \right) \\
&\leq R/2 + R/20 + R/8 + R/8 \\
&< R.
\end{aligned}
$$

This a contradiction and thus for all time $s \leq t$, we have that

$$
\sqrt{\|U_s - U_0\|^2 + \alpha \|V_s - V_0\|^2} < R
$$

By eq. (66) and since $R = \frac{\gamma^{1/3} m^{1/6}}{27 \epsilon_N}$, we get

$$
\sum_{s<t} \sum_{i \in [n]} \frac{1}{n} \left| \ell_i'(W_s) \right| \leq \frac{4\|U_\tau - U_0\|}{\eta \gamma \sqrt{m}} \leq \frac{4R}{\eta \gamma \sqrt{m}} \leq \frac{2^{22} \epsilon_N^4}{\gamma^{1/3}}.
$$

Dividing both sides by t, we obtain

$$\inf_{s \in [0,t]} \sum_{i \in [n]} \frac{-\ell_i'(W_s)}{n} \le \frac{2^{22} m^{1/3} \epsilon_N^4}{\gamma^{1/3} t}.$$

Let $\rho \in (0, 1)$. We will first show that $g(z) = -\ell'(z)$ is $\rho$-lipschitz on the interval $[g^{-1}(\rho), \infty)$. To show this, we will show the uniform bound $|g'| \le \rho$ on the interval $[g^{-1}(\rho), \infty)$. Note that

$$|g'(z)| = \frac{\exp(z)}{\left(1 + \exp(z)\right)^2}.$$

which is monotonically decreasing on the interval $[g^{-1}(\rho), \infty)$. Therefore, $|g'(z)|$ is the largest when $z = g^{-1}(\rho)$. Since $g^{-1}(z) = \ln(\frac{1}{z} - 1)$,

$$\sup_{z \in [g^{-1}(\rho), \infty)} |g'(z)| = |g'(g^{-1}(\rho))| = \frac{\rho^{-1} - 1}{\rho^{-2}} \le \rho. \tag{68}$$

For the remainder of the proof, let $\rho = \frac{4R}{\gamma\sqrt{m}}$. Now define our hypothesis class as

$$\mathcal{H} = \left\{ (x, y) \to y \sum_{j \in [m]} a_j \sigma(c_j \left\langle \frac{v_j}{N(v_j)}, x - \mu \right\rangle) : \ W \in \Gamma, \max_i -\ell'(p_i(W)) \le \rho \right\}.$$

Since $g \in [0, 1]$, we can apply Shalev-Shwartz & Ben-David (2014, theorem 26.3) to get, with probability at least $1 - \delta$,

$$\sup_{h \in \mathcal{H}} \left( \mathbb{E}_{z=(x,y)} \left[ g(h(z)) \right] - \frac{1}{n} \sum_{i \in [n]} g(h(z_i)) \right) \le 2\text{Rad}(g \circ \mathcal{H} * \mathcal{X}) + 3\sqrt{\frac{\ln(2/\delta)}{2n}}. \tag{69}$$

To control $2\text{Rad}(g \circ \mathcal{H} * \mathcal{X})$, we first note that $\mathcal{H} \circ \mathcal{X} \subset [g^{-1}(\rho), \infty)^d$. Furthermore, $g$ is $\rho$-lipschitz on $[g^{-1}(\rho), \infty)$ by eq. (68) and thus we can invoke lemma F.2 to get

$$\text{Rad}(g \circ \mathcal{H} * \mathcal{X}) \le \rho\text{Rad}(\mathcal{H} * \mathcal{X}).$$

We shall bound $\text{Rad}(\mathcal{H} * \mathcal{X})$ by invoking lemma F.1. To this end, first note that for any $W' \in \Gamma$,

$$\|a'\| \le \|a' - a_0\| + \|a_0\| \le R + 2 \le 10,$$
$$\|c'\| \le \|c' - c_0\| + \|c_0\| \le R + \sqrt{m} \le 2\sqrt{m}.$$

Therefore, for any $W' \in \Gamma$

$$\sum_{j \in [m]} \frac{|a_j' c_j'|}{N(v_j')} \le \|a'\|\|c'\| \epsilon_N \le 20\epsilon_N \sqrt{m}.$$

Furthermore, $W' \in \Gamma$ implies $\|V' - V\| < R_V = \frac{1}{2\epsilon_N}$. Therefore, by lemma F.1, we have

$$\text{Rad}(\mathcal{H} * \mathcal{X}) \le \frac{40\sqrt{m}}{\sqrt{n}}.$$

Thus, recalling that $\rho = \frac{4R}{\gamma\sqrt{m}} = \frac{32}{\gamma\sqrt{m}}$, eq. (69) simplifies to

$$\sup_{h \in \mathcal{H}} \left( \mathbb{E}_{z=(x,y)} \left[ g(h(z)) \right] - \frac{1}{n} \sum_{i \in [n]} g(h(z_i)) \right) \le \frac{80\sqrt{m}\rho}{\sqrt{n}} + 3\sqrt{\frac{\ln(2/\delta)}{2n}} \tag{70}$$

$$\le \frac{2560}{\gamma\sqrt{n}} + 3\sqrt{\frac{\ln(2/\delta)}{2n}}. \tag{71}$$

Finally, using the fact that $\Pr(y \neq f(x; W)) \leq 2\mathbb{E}_{z=(x,y)}\big[g(yf(x; W))\big]$ and rearranging eq. (70),

$$\inf_{s \leq t} \Pr(y \neq f(x; W_t)) \leq \inf_{s \leq t} 2\mathbb{E}_{z=(x,y)}\big[g(yf(x; W_t))\big]$$

$$\leq \frac{2}{n} \sum_{i \in [n]} g(y_i f(x_i; W_t)) + \frac{2560}{\gamma\sqrt{n}} + 3\sqrt{\frac{\ln(2/\delta)}{2n}}$$

$$\leq \frac{32}{\gamma n\sqrt{m}} + \frac{2560}{\gamma\sqrt{n}} + 3\sqrt{\frac{\ln(2/\delta)}{2n}}$$

$$\leq \frac{2^{11}}{\gamma\sqrt{n}} + 3\sqrt{\frac{\ln(2/\delta)}{2n}}.$$

$\square$

# G   COMPUTATION OF BN MARGIN ON DYADIC DATASET

In this section, we provide a proof of proposition 1.1.

*Proof.* Recall the dyadic dataset. Let $e_1, \ldots, e_d \in \mathbb{R}^d$ denote the standard basis vectors.

$$(x_j, y_j) = (2^{-j}e_j, +1), \quad j \in \{1, \ldots, d\}$$
$$(x_j, y_j) = (-2^{-j+d}e_{j-d}, -1), \quad j \in \{d+1, \ldots, n\}$$

Then the covariance matrix $\Sigma$ satisfies

$$\Sigma_{ij} = \begin{cases} 0 & i \neq j \\ \frac{2^{-2j}}{d} & i = j \end{cases}$$

Now consider the vector $u = (2, 4, \ldots, 2^d)$ and define the unit vector $\vec{u} := \frac{u}{\|u\|}$. We note that $\vec{u}$ is the max margin linear predictor for the dataset. Furthermore, the margin satisfies

$$\gamma_{\text{linear}} := \min_{i \in [2d]} y_i x_i^\mathsf{T}\vec{u} = \frac{1}{\sqrt{\sum_{r=1}^d 2^{2r}}} = \frac{1}{\sqrt{\frac{4^{d+1}-1}{4-1} - 1}} = \sqrt{\frac{3}{4^{d+1}-4}} \leq \sqrt{\frac{3}{4^{d+1}}}.$$

Consider the infinite network

$$x \to \mathbb{E}[\text{sgn}(\vec{u}^\mathsf{T}v)\sigma(\frac{v^\mathsf{T}x}{\|v\|_\Sigma})].$$

Let M be the orthonormal matrix $M \in \mathbb{R}^{d \times d}$ such that its first column is $\vec{u}$ and the second column is $\frac{(I - \vec{u}\vec{u}^\mathsf{T})x}{\|(I - \vec{u}\vec{u}^\mathsf{T})x\|}$, and the remaining columns be any vectors such that $M$ is orthonormal. Consequently, letting $r_2 = \sqrt{\|x\|^2 - (\vec{u}^\mathsf{T})^2}$, matrix $M$ satisfies

$$M\vec{u} = e_1 \quad Mx = e_1\vec{u}^\mathsf{T}x + r_2 e_2. \tag{72}$$

By rotational invariance of gaussians, eq. (72), and since $\|Mv\| = \|v\|$,

$$\mathbb{E}[y\,\text{sign}(\vec{u}^\mathsf{T}v)\sigma(\frac{v^\mathsf{T}x}{\|v\|_\Sigma})] = \mathbb{E}[y\text{sgn}(v_1)\sigma(\frac{v\vec{u}^\mathsf{T}xy^2 + v_2 r_2}{\|Mv\|_\Sigma})] \tag{73}$$

$$= \mathbb{E}_{\substack{y\text{sgn}(v_1)=1 \\ v_2 \geq 0}}\Big[\frac{\sigma(|v_1|\,\vec{u}^\mathsf{T}xy + v_2 r_2) - \sigma(|v_1|\,\vec{u}^\mathsf{T}xy + v_2 r_2)}{\|Mv\|_\Sigma}\Big] \tag{74}$$

$$+ \mathbb{E}_{\substack{y\text{sgn}(v_1)=1 \\ v_2 \geq 0}}\Big[\frac{\sigma(|v_1|\,\vec{u}^\mathsf{T}xy - v_2 r_2) - \sigma(-|v_1|\,\vec{u}^\mathsf{T}xy - v_2 r_2)}{\|Mv\|_\Sigma}\Big]. \tag{75}$$

Inspecting the signs of the ReLU argument, we see that the first ReLu argument is positive, exactly one of the second and third argument is postive, and the fourth argument is negative. Therefore, the eq. (73) simplifies to

$$\mathbb{E}[y \operatorname{sign}(\overrightarrow{u}^{\mathsf{T}} v)\sigma(\frac{v^{\mathsf{T}} x}{\|v\|_{\Sigma}})] = 2\mathbb{E}_{\substack{y \operatorname{sgn}(v_1)=1 \\ v_2 \geq 0}}[\frac{|v_1| \overrightarrow{u}^{\mathsf{T}} xy}{\|Mv\|_{\Sigma}}] = \frac{1}{2}\mathbb{E}[\frac{|v_1| \overrightarrow{u}^{\mathsf{T}} xy}{\|Mv\|_{\Sigma}}].$$

Finally, noting that $\overrightarrow{u}^{\mathsf{T}} xy \geq \gamma_{\text{linear}}$ and $\mathbb{E}[\frac{|v_1|}{\|Mv\|_{\Sigma}}] \geq \frac{\sqrt{d}}{10}$ grants

$$\mathbb{E}[y \operatorname{sign}(\overrightarrow{u}^{\mathsf{T}} v)\sigma(\frac{v^{\mathsf{T}} x}{\|v\|_{\Sigma}})] \geq \frac{1}{20}\gamma_{\text{linear}}.$$

$\square$

