# OpenReview forum: "Test Error Guarantees for Batch-normalized two-layer ReLU Networks Trained with Gradient Descent"
_ICLR.cc/2024/Conference — ICLR 2024 Conference Withdrawn Submission_

### Official Review · Reviewer_ryx4 · 2023-10-31

**Soundness:** 2 fair
**Presentation:** 4 excellent
**Contribution:** 3 good
**Rating:** 5
**Confidence:** 4

**Summary:**

This paper proves the low training (with logistic loss) and test error guarantees of GD/SGD for two-layer ReLU neural network with batch normalization. It mainly uses the techniques developed in Telgarsky(2022) and Ji & Telgarsky(2020) to show that the weights will move closer to the scaled gradient of the initial predictor ($df/dw_0$) then the margin can be lower bounded then the training and test error can be bounded.

**Strengths:**

1. It is a novel theoretical result.  It keeps the $\epsilon$ in the variance term in batch norm in contrast to Arora et al. (2018).  Hence the analysis is closer to the batch norm used in practice.

2. I like the idea of rescaled Euclidean potential which allows separate control over different parameters. I think it can be potentially applied to other analyses.

**Weaknesses:**

1. My major concern is about the size of the bound in Theorem 2.1. The bound on $m$ is too large as it has constant $2^{29}$. And in Eq.(2) there is a term $2^{15}$. Even for a theory paper, I think the constants are excessively high.  I would suggest a re-evaluation of these terms to achieve more reasonable bounds.

2. I believe the intuition for Assumption 1.1 should be further clarified, e.g., how does it affect the proof?

3. In Proposition 1.1, the chosen parameters are too special ($C = 0$) since $f(x) = 0$ for any input x.  I don't think GD/SGD will converge to such parameters.

**Questions:**

1. Why do the bounds on smoothness can be as large as $O(M^L)$? At least in the NTK setting, the bounds shouldn’t grow with the width and the number of layers. I believe in other practical settings, the bounds shouldn't be that large as well. Why do you think such an assumption is problematic?

2. Is there an error in the proof of Theorem 2.1 (F.4)?  Before Eq.(66), you used the fact that $||\overline{U}||\leq 1$. I didn't find the support for this fact.

3. In the proof of Lemma B.1., it seems like $\epsilon$ is assumed to be positive. Then I am confused about the sentence after Assumption 1.1. Could you clarify?

---

### Official Review · Reviewer_J2f7 · 2023-11-01

**Soundness:** 3 good
**Presentation:** 2 fair
**Contribution:** 2 fair
**Rating:** 5
**Confidence:** 2

**Summary:**

Authors consider gradient descent (GD) and stochastic gradient descent (SGD) on two-layer ReLU networks with
Batch Norm. Consider the test error rate with respect to a parameter $gamma$ similar to a parameter used by Telgarsky (2022) previously. They show that the test error decreases at a rate of $O(\frac{m^{1/3}}{\gamma^{1/3}t})$ and $ O(\frac{1}{\gamma^2 t})$ for networks with width $\Omega(1/\gamma^2)$.

**Strengths:**

-Combines two original ideas to understand and control (two layer ReLU) neural networks, batch normalized networks and the margin introduced by Ji & Telgarsky (2020).

-mathematical proofs are provided as well as experiments (though I did not check everything for correctness)

**Weaknesses:**

-write up could use some improvements (some definitions are missing,for example what is sigma what is U, there are lots of definitions and it is hard to follow the paper, sometimes the order is confusing and not everything is explained well, for example Assumption 1.1 could follow immeadialtely after the definiton of N(v) and both could be explained in more detail why they are used)

-very similar to previous papers

**Questions:**

-

---

### Official Review · Reviewer_4dGx · 2023-11-04

**Soundness:** 3 good
**Presentation:** 2 fair
**Contribution:** 2 fair
**Rating:** 5
**Confidence:** 4

**Summary:**

This paper considers the two-layer ReLU networks with batch normalization (BN). In particular, this paper proposes a new margin definition for the two-layer ReLU network with BN, and develops the test error bound of SGD and GD using the proposed margin definitions.  The results can be utilized to better understand the performance of BN and provide a new method for evaluating the generalization performance.

**Strengths:**

* A new definition of margin that can be useful to better evaluate the generalization performance.
* A generalization bound for SGD/GD on batch-normalized 2-layer networks.

**Weaknesses:**

1. The paper's organization is not clear. For instance, in Section 1.4, I just quite not understand that you first say $\bar W = W_0 + r\overset{\rightarrow}{W}$ is difficult to use and then mention that we again consider this $\bar W$.
2. I am not sure whether only considering the optimization around the initialization is good as this is different from the practice.
3. After reading the paper, I am not sure what role the normalization layer plays in the analysis since the authors only consider the margin with respect to the scaling parameters $a$ and $c$. While the hidden layer weights $v$ could be more important.
4. I also cannot quite understand why you need to ensure that $\|v_i/N(v_i)-v_j/N(v_j)\|$ is small.
5. The new margin definition is also weird as it's written as an assumption rather than a definition. So would it be possible that the positive $\gamma$ does not exist?
6. In proposition 1.1, I do not think that the comparison of different margins is fair as they are defined in different ways. On the other hand, can you also give the test error bound for the model without a normalization layer?
7. The SGD case is also confusing. To my understanding, SGD with BN only estimates the data covariance using a random mini-batch, so that the expectation of stochastic gradient will not be the full gradient. Then I do not see why can you use a similar analysis to get the theoretical results for both SGD and GD?
8. What if we only consider linear separable data, then what's the relationship between the margin defined in this paper and the margin of the data? When using your definition of margin, what's the new margin of the original max-margin classifier? Besides, a very recent paper [1] shows that the BN could lead to a ``uniform margin'' solution (defined using the conventional margin definition), would it be also possible to translate such a margin using your definition?

[1 ] Cao et al., The Implicit Bias of Batch Normalization in Linear Models and Two-layer Linear Convolutional Neural Networks, COLT 2023

**Questions:**

See the weakness part.

---

### Official Review · Reviewer_jFN6 · 2023-11-08

**Soundness:** 1 poor
**Presentation:** 3 good
**Contribution:** 3 good
**Rating:** 3
**Confidence:** 5

**Summary:**

This paper aims to show that online SGD (with fresh sample at each step) or full-batch GD (on finite dataset) provably decrease test errors of two-layer ReLU net with Batch Normalization to 0 as t -> inf.

**Strengths:**

1. This paper is well-written. Theoretical setups and assumptions are presented very clearly.
2. This paper looks into the training dynamics of neural nets with normalization layers, which is an interesting and challenging case to analyze.

**Weaknesses:**

**Main Concern:** This paper would be interesting if it really presents a new analysis that goes beyond the previous works mentioned in the introduction, but it is unclear to me whether the proof is indeed correct.
1. For both the SGD and GD cases, it is shown that the parameters U will be trapped in a neighborhood of their initial values, and it is implied from the proof that the training loss will go to 0 as t -> inf (since the sum of $|\ell'|$ over time is finite). However, these two cannot happen at the same time: if U is trapped in a finite region, then the output of the neural net is also finite, and thus, the logistic loss must be lower bounded by a positive constant. To make it 0, the output as well as the parameters in U has to go to infinity.
2. The result of GD implies that 0 test error can be obtained even if there is only 1 data point. To see this, just take t -> inf in equation (6). Obviously, having only 1 data point is not information-theoretically possible to learn a perfect classifier.

**A minor issue:** Page 7, $\Gamma$ is just a set of points, not a manifold.

**Questions:**

I urge the authors to check the correctness of the proof.